# FEW-SHOT LEARNING AS CLUSTER-INDUCED VORONOI DIAGRAMS: A GEOMETRIC APPROACH

**Chunwei Ma**[1], **Ziyun Huang**[2], **Mingchen Gao**[1], **Jinhui Xu**[1]

[1]Department of Computer Science and Engineering, University at Buffalo
[2]Computer Science and Software Engineering, Penn State Erie
[1]{chunweim,mgao8,jinhui}@buffalo.edu
[2]{zxh201}@psu.edu

## ABSTRACT

Few-shot learning (FSL) is the process of rapid generalization from abundant base samples to inadequate novel samples. Despite extensive research in recent years, FSL is still not yet able to generate satisfactory solutions for a wide range of real-world applications. To confront this challenge, we study the FSL problem from a geometric point of view in this paper. One observation is that the widely embraced ProtoNet model is essentially a Voronoi Diagram (VD) in the feature space. We retrofit it by making use of a recent advance in computational geometry called *Cluster-induced Voronoi Diagram (CIVD)*. Starting from the simplest nearest neighbor model, CIVD gradually incorporates cluster-to-point and then cluster-to-cluster relationships for space subdivision, which is used to improve the accuracy and robustness at multiple stages of FSL. Specifically, we use CIVD (1) to integrate parametric and nonparametric few-shot classifiers; (2) to combine feature representation and surrogate representation; (3) and to leverage feature-level, transformation-level, and geometry-level heterogeneities for a better ensemble. Our CIVD-based workflow enables us to achieve new state-of-the-art results on mini-ImageNet, CUB, and tiered-ImagenNet datasets, with $\sim 2\%-5\%$ improvements upon the next best. To summarize, CIVD provides a mathematically elegant and geometrically interpretable framework that compensates for extreme data insufficiency, prevents overfitting, and allows for fast geometric ensemble for thousands of individual VD. These together make FSL stronger.

## 1 INTRODUCTION

Recent years have witnessed a tremendous success of deep learning in a number of data-intensive applications; one critical reason for which is the vast collection of hand-annotated high-quality data, such as the millions of natural images for visual object recognition (Deng et al., 2009). However, in many real-world applications, such large-scale data acquisition might be difficult and comes at a premium, such as in rare disease diagnosis (Yoo et al., 2021) and drug discovery (Ma et al., 2021b; 2018). As a consequence, Few-shot Learning (FSL) has recently drawn growing interests (Wang et al., 2020).

Generally, few-shot learning algorithms can be categorized into two types, namely *inductive* and *transductive*, depending on whether estimating the distribution of query samples is allowed. A typical transductive FSL algorithm learns to propagate labels among a larger pool of query samples in a semi-supervised manner (Liu et al., 2019); notwithstanding its normally higher performance, in many real world scenarios a query sample (e.g. patient) also comes individually and is unique, for instance, in personalized pharmacogenomics (Sharifi-Noghabi et al., 2020). Thus, we in this paper adhere to the inductive setting and make on-the-fly prediction for each newly seen sample.

Few-shot learning is challenging and substantially different from conventional deep learning, and has been tackled by many researchers from a wide variety of angles. Despite the extensive research

---

All four authors are corresponding authors.

on the algorithmic aspects of FSL (see Sec. 2), two challenges still pose an obstacle to successful FSL: (1) how to sufficiently compensate for the data deficiency in FSL? and (2) how to make the most use of the base samples and the pre-trained model?

For the first question, data augmentation has been a successful approach to expand the size of data, either by Generative Adversarial Networks (GANs) (Goodfellow et al., 2014) (Li et al., 2020b; Zhang et al., 2018) or by variational autoencoders (VAEs) (Kingma & Welling, 2014) (Zhang et al., 2019; Chen et al., 2019b). However, in each way, the authenticity of either the augmented data or the feature is not guaranteed, and the out-of-distribution hallucinated samples (Ma et al., 2019) may hinder the subsequent FSL. Recently, Liu et al. (2020b) and Ni et al. (2021) investigate support-level, query-level, task-level, and shot-level augmentation for meta-learning, but the diversity of FSL models has not been taken into consideration. For the second question, Yang et al. (2021) borrows the top-2 nearest base classes for each novel sample to calibrate its distribution and to generate more novel samples. However, when there is no proximal base class, this calibration may utterly alter the distribution. Another line of work (Sbai et al., 2020; Zhou et al., 2020) learns to select and design base classes for a better discrimination on novel classes, which all introduce extra training burden. As a matter of fact, we still lack a method that makes full use of the base classes and the pretrained model effectively.

In this paper, we study the FSL problem from a geometric point of view. In metric-based FSL, despite being surprisingly simple, the nearest neighbor-like approaches, e.g. ProtoNet (Snell et al., 2017) and SimpleShot (Wang et al., 2019), have achieved remarkable performance that is even better than many sophisticatedly designed methods. Geometrically, what a nearest neighbor-based method does, under the hood, is to partition the feature space into a *Voronoi Diagram (VD)* that is induced by the feature centroids of the novel classes. Although it is highly efficient and simple, Voronoi Diagrams coarsely draw the decision boundary by linear bisectors separating two centers, and may lack the ability to subtly delineate the geometric structure arises in FSL.

To resolve this issue, we adopt a novel technique called *Cluster-induced Voronoi Diagram (CIVD)* (Chen et al., 2013; 2017; Huang & Xu, 2020; Huang et al., 2021), which is a recent breakthrough in computation geometry. CIVD generalizes VD from a point-to-point distance-based diagram to a cluster-to-point influence-based structure. It enables us to determine the dominating

Table 1: The underlying geometric structures for various FSL methods.

| Method | Geometric Structure |
|---|---|
| ProtoNet (Snell et al., 2017) | Voronoi Diagram |
| S2M2_R (Mangla et al., 2020) | spherical VD |
| DC (Yang et al., 2021) | Power Diagram |
| DeepVoro−− (ours) | CIVD |
| DeepVoro/DeepVoro++ (ours) | CCVD |

region (or Voronoi cell) not only for a point (e.g. a class prototype) but also for a *cluster* of points, guaranteed to have a $(1 + \epsilon)$-approximation with a nearly linear size of diagram for a wide range of locally dominating influence functions. CIVD provides us a mathematically elegant framework to depict the feature space and draw the decision boundary more precisely than VD without losing the resistance to overfitting.

Accordingly, in this paper, we show how CIVD is used to improve multiple stages of FSL and make several contributions as follows.

**1.** We first categorize different types of few-shot classifiers as different variants of Voronoi Diagram: nearest neighbor model as Voronoi Diagram, linear classifier as Power Diagram, and cosine classifier as spherical Voronoi Diagram (Table 1). We then unify them via CIVD that enjoys the advantages of multiple models, either parametric or nonparametric (denoted as DeepVoro−−).

**2.** Going from cluster-to-point to cluster-to-cluster influence, we further propose *Cluster-to-cluster Voronoi Diagram (CCVD)*, as a natural extension of CIVD. Based on CCVD, we present DeepVoro which enables fast geometric ensemble of a large pool of thousands of configurations for FSL.

**3.** Instead of using base classes for distribution calibration and data augmentation (Yang et al., 2021), we propose a novel *surrogate representation*, the collection of similarities to base classes, and thus promote DeepVoro to DeepVoro++ that integrates feature-level, transformation-level, and geometry-level heterogeneities in FSL.

Extensive experiments have shown that, although a fixed feature extractor is used without independently pretrained or epoch-wise models, our method achieves new state-of-the-art results on all

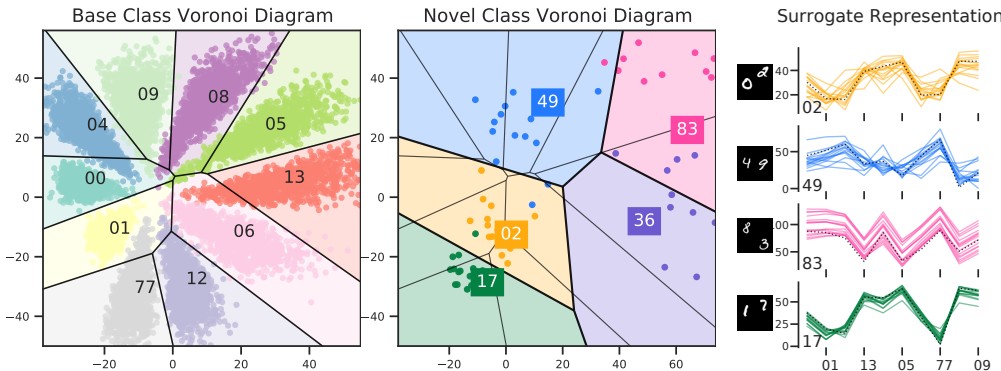

Figure 1: Schematic illustrations of Voronoi Diagram (VD) and surrogate representation on Multi-DigitMNIST dataset (Sun, 2019). Left and central panels demonstrate the VD of base classes and novel classes (5-way 1-shot) in $\mathbb{R}^2$, respectively. The colored squares stand for the 1-shot support samples. In the right panel, for each support sample, the surrogate representation (dotted line) exhibits a unique pattern which those of the query samples (colored lines) also follow. (See Appendix C for details.)

three benchmark datasets including mini-ImageNet, CUB, and tiered-ImageNet, and improves by up to $2.18\%$ on 5-shot classification, $2.53\%$ on 1-shot classification, and up to $5.55\%$ with different network architectures.

## 2 RELATED WORK

**Few-Shot Learning.** There are a number of different lines of research dedicated to FSL. (1) Metric-based methods employ a certain distance function (cosine distance (Mangla et al., 2020; Xu et al., 2021), Euclidean distance (Wang et al., 2019; Snell et al., 2017), or Earth Mover's Distance (Zhang et al., 2020a;b)) to bypass the optimization and avoid possible overfitting. (2) Optimization-based approaches (Finn et al., 2017) manages to learn a good model initialization that accelerates the optimization in the meta-testing stage. (3) Self-supervised-based (Zhang et al., 2021b; Mangla et al., 2020) methods incorporate supervision from data itself to learn a robuster feature extractor. (4) Ensemble method is another powerful technique that boosting the performance by integrating multiple models (Ma et al., 2021a). For example, Dvornik et al. (2019) trains several networks simultaneously and encourages robustness and cooperation among them. However, due to the high computation load of training deep models, this ensemble is restricted by the number of networks which is typically <20. In Liu et al. (2020c), instead, the ensemble consists of models learned at each epoch, which, may potentially limit the diversity of ensemble members.

**Geometric Understanding of Deep Learning.** The geometric structure of deep neural networks is first hinted at by Raghu et al. (2017) who reveals that piecewise linear activations subdivide input space into convex polytopes. Then, Balestriero et al. (2019) points out that the exact structure is a Power Diagram (Aurenhammer, 1987) which is subsequently applied upon recurrent neural network (Wang et al., 2018) and generative model (Balestriero et al., 2020). The Power/Voronoi Diagram subdivision, however, is not necessarily the optimal model for describing feature space. Recently, Chen et al. (2013; 2017); Huang et al. (2021) uses an influence function $F(\mathcal{C}, z)$ to measure the joint influence of all objects in $\mathcal{C}$ on a query $z$ to build a Cluster-induced Voronoi Diagram (CIVD). In this paper, we utilize CIVD to magnify the expressivity of geometric modeling for FSL.

## 3 METHODOLOGY

### 3.1 PRELIMINARIES

Few-shot learning aims at discriminating between novel classes $\mathcal{C}^{\text{novel}}$ with the aid of a larger amount of samples from base classes $\mathcal{C}^{\text{base}}, \mathcal{C}^{\text{novel}} \cap \mathcal{C}^{\text{base}} = \emptyset$. The whole learning process usually follows the

*meta-learning* scheme. Formally, given a dataset of base classes $\mathcal{D} = \{(\boldsymbol{x}_i, y_i)\}, \boldsymbol{x}_i \in \mathbb{D}, y_i \in \mathcal{C}^{\text{base}}$ with $\mathbb{D}$ being an arbitrary domain e.g. natural image, a deep neural network $\boldsymbol{z} = \phi(\boldsymbol{x}), \boldsymbol{z} \in \mathbb{R}^n$, which maps from image domain $\mathbb{D}$ to feature domain $\mathbb{R}^n$, is trained using standard gradient descent algorithm, and after which $\phi$ is fixed as a feature extractor. This process is referred to as *meta-training* stage that squeezes out the commonsense knowledge from $\mathcal{D}$.

For a fair evaluation of the learning performance on a few samples, the *meta-testing* stage is typically formulated as a series of $K$-way $N$-shot tasks (episodes) $\{\mathcal{T}\}$. Each such episode is further decomposed into a *support set* $\mathcal{S} = \{(\boldsymbol{x}_i, y_i)\}_{i=1}^{K \times N}, y_i \in \mathcal{C}^{\mathcal{T}}$ and a *query set* $\mathcal{Q} = \{(\boldsymbol{x}_i, y_i)\}_{i=1}^{K \times Q}, y_i \in \mathcal{C}^{\mathcal{T}}$, in which the episode classes $\mathcal{C}^{\mathcal{T}}$ is a randomly sampled subset of $\mathcal{C}^{\text{novel}}$ with cardinality $K$, and each class contains only $N$ and $Q$ random samples in the support set and query set, respectively. For few-shot classification, we introduce here two widely used schemes as follows. For simplicity, all samples here are from $\mathcal{S}$ and $\mathcal{Q}$, without data augmentation applied.

**Nearest Neighbor Classifier (Nonparametric).** In Snell et al. (2017); Wang et al. (2019) etc., a *prototype* $\boldsymbol{c}_k$ is acquired by averaging over all supporting features for a class $k \in \mathcal{C}^{\mathcal{T}}$:

$$\boldsymbol{c}_k = \frac{1}{N} \sum_{\boldsymbol{x} \in \mathcal{S}, y=k} \phi(\boldsymbol{x}) \tag{1}$$

Then each query sample $\boldsymbol{x} \in \mathcal{Q}$ is classified by finding the nearest prototype: $\hat{y} = \arg\min_k d(\boldsymbol{z}, \boldsymbol{c}_k) = ||\boldsymbol{z} - \boldsymbol{c}_k||_2^2$, in which we use Euclidean distance for distance metric $d$.

**Linear Classifier (Parametric).** Another scheme uses a linear classifier with cross-entropy loss optimized on the supporting samples:

$$\mathcal{L}(\boldsymbol{W}, \boldsymbol{b}) = \sum_{(\boldsymbol{x},y) \in \mathcal{S}} - \log p(y|\phi(\boldsymbol{x}); \boldsymbol{W}, \boldsymbol{b}) = \sum_{(\boldsymbol{x},y) \in \mathcal{S}} - \log \frac{\exp(\boldsymbol{W}_y^T \phi(\boldsymbol{x}) + b_y)}{\sum_k \exp(\boldsymbol{W}_k^T \phi(\boldsymbol{x}) + b_k)} \tag{2}$$

in which $\boldsymbol{W}_k, b_k$ are the linear weight and bias for class $k$, and the predicted class for query $\boldsymbol{x} \in \mathcal{Q}$ is $\hat{y} = \arg\max_k p(y|\boldsymbol{z}; \boldsymbol{W}_k, b_k)$.

## 3.2 FEW-SHOT LEARNING AS CLUSTER-INDUCED VORONOI DIAGRAMS

In this section, we first introduce the basic concepts of Voronoi Tessellations, and then show how parametric/nonparametric classifier heads can be unified by VD.

**Definition 3.1** (Power Diagram and Voronoi Diagram). Let $\Omega = \{\omega_1, ..., \omega_K\}$ be a partition of the space $\mathbb{R}^n$, and $\mathcal{C} = \{\boldsymbol{c}_1, ..., \boldsymbol{c}_K\}$ be a set of centers such that $\cup_{r=1}^K \omega_r = \mathbb{R}^n, \cap_{r=1}^K \omega_r = \emptyset$. Additionally, each center is associated with a weight $\nu_r \in \{\nu_1, ..., \nu_K\} \subseteq \mathbb{R}^+$. Then the set of pairs $\{(\omega_1, \boldsymbol{c}_1, \nu_1), ..., (\omega_K, \boldsymbol{c}_L, \nu_K)\}$ is a Power Diagram (PD), where each cell is obtained via $\omega_r = \{\boldsymbol{z} \in \mathbb{R}^n : r(\boldsymbol{z}) = r\}, r \in \{1, .., K\}$, with

$$r(\boldsymbol{z}) = \underset{k \in \{1, ..., K\}}{\arg\min} \; d(\boldsymbol{z}, \boldsymbol{c}_k)^2 - \nu_k. \tag{3}$$

If the weights are equal for all $k$, i.e. $\nu_k = \nu_{k'}, \forall k, k' \in \{1, ..., K\}$, then a PD collapses to a Voronoi Diagram (VD).

By definition, it is easy to see that the nearest neighbor classifier naturally partitions the space into $K$ cells with centers $\{\boldsymbol{c}_1, ..., \boldsymbol{c}_K\}$. Here we show that the linear classifier is also a VD under a mild condition.

**Theorem 3.1** (Voronoi Diagram Reduction). *The linear classifier parameterized by $\boldsymbol{W}, \boldsymbol{b}$ partitions the input space $\mathbb{R}^n$ to a Voronoi Diagram with centers $\{\tilde{\boldsymbol{c}}_1, ..., \tilde{\boldsymbol{c}}_K\}$ given by $\tilde{\boldsymbol{c}}_k = \frac{1}{2}\boldsymbol{W}_k$ if $b_k = -\frac{1}{4}||\boldsymbol{W}_k||_2^2, k = 1, ..., K$.*

*Proof.* See Appendix B for details. □

### 3.2.1 FROM VORONOI DIAGRAM TO CLUSTER-INDUCED VORONOI DIAGRAM

Now that both nearest neighbor and linear classifier have been unified by VD, a natural idea is to integrate them together. Cluster-induced Voronoi Diagram (CIVD) (Chen et al., 2017; Huang et al.,

2021) is a generalization of VD which allows multiple centers in a cell, and is successfully used for clinical diagnosis from biomedical images (Wang et al., 2015), providing an ideal tool for the integration of parametric/nonparametric classifier for FSL. Formally:

**Definition 3.2** (Cluster-induced Voronoi Diagram (CIVD) (Chen et al., 2017; Huang et al., 2021)). Let $\Omega = \{\omega_1, ..., \omega_K\}$ be a partition of the space $\mathbb{R}^n$, and $\mathcal{C} = \{\mathcal{C}_1, ..., \mathcal{C}_K\}$ be a set (possibly a multiset) of *clusters*. The set of pairs $\{(\omega_1, \mathcal{C}_1), ..., (\omega_K, \mathcal{C}_K)\}$ is a Cluster-induced Voronoi Diagram (CIVD) with respect to the influence function $F(\mathcal{C}_k, \boldsymbol{z})$, where each cell is obtained via $\omega_r = \{\boldsymbol{z} \in \mathbb{R}^n : r(\boldsymbol{z}) = r\}, r \in \{1, .., K\}$, with

$$r(\boldsymbol{z}) = \underset{k \in \{1, ..., K\}}{\arg\max} \; F(\mathcal{C}_k, \boldsymbol{z}). \tag{4}$$

Here $\mathcal{C}$ can be either a given set of clusters or even the whole power set of a given point set, and the influence function is defined as a function over the collection of distances from each member in a cluster $\mathcal{C}_k$ to a query point $\boldsymbol{z}$:

**Definition 3.3** (Influence Function). The influence from $\mathcal{C}_k, k \in \{1, ..., K\}$ to $\boldsymbol{z} \notin \mathcal{C}_k$ is $F(\mathcal{C}_k, \boldsymbol{z}) = F(\{d(\boldsymbol{c}_k^{(i)}, \boldsymbol{z}) | \boldsymbol{c}_k^{(i)} \in \mathcal{C}_k\}_{i=1}^{|\mathcal{C}_k|})$. In this paper $F$ is assumed to have the following form

$$F(\mathcal{C}_k, \boldsymbol{z}) = -\operatorname{sign}(\alpha) \sum_{i=0}^{|\mathcal{C}_k|} d(\boldsymbol{c}_k^{(i)}, \boldsymbol{z})^\alpha. \tag{5}$$

The sign function here makes sure that $F$ is a monotonically decreasing function with respect to distance $d$. The hyperparameter $\alpha$ controls the magnitude of the influence, for example, in gravity force $\alpha = -(n-1)$ in $n$-dimensional space and in electric force $\alpha = -2$.

Since the nearest neighbor centers $\{\boldsymbol{c}_k\}_{k=1}^K$ and the centers introduced by linear classifier $\{\tilde{\boldsymbol{c}}_k\}_{k=1}^K$ are obtained from different schemes and could both be informative, we merge the corresponding centers for a novel class $k$ to be a new cluster $\mathcal{C}_k = \{\boldsymbol{c}_k, \tilde{\boldsymbol{c}}_k\}$, and use the resulting $\mathcal{C} = \{\mathcal{C}_1, ..., \mathcal{C}_K\}$ to establish a CIVD. In such a way, the final partition may enjoy the advantages of both parametric and nonparametric classifier heads. We name this approach as DeepVoro−−.

### 3.3 FEW-SHOT CLASSIFICATION VIA SURROGATE REPRESENTATION

In nearest neighbor classifier head, the distance from a query feature $\boldsymbol{z}$ to each of the prototypes $\{\boldsymbol{c}_k\}_{k=1}^K$ is the key discrimination criterion for classification. We rewrite $\{d(\boldsymbol{z}, \boldsymbol{c}_k)\}_{k=1}^K$ to be a vector $\boldsymbol{d} \in \mathbb{R}^K$ such that $d_k = d(\boldsymbol{z}, \boldsymbol{c}_k)$. These distances are acquired by measure the distance between two points in high dimension: $\boldsymbol{z}, \boldsymbol{c}_k \in \mathbb{R}^n$. However, the notorious behavior of high dimension is that the ratio between the nearest and farthest points in a point set $\mathcal{P}$ approaches 1 (Aggarwal et al., 2001), making $\{d(\boldsymbol{z}, \boldsymbol{c}_k)\}_{k=1}^K$ less discriminative for classification, especially for FSL problem with sample size $N \cdot K \ll n$. Hence, in this paper, we seek for a *surrogate representation*.

In human perception and learning system, similarity among *familiar* and *unfamiliar* objects play a key role for object categorization and classification (Murray et al., 2002), and it has been experimentally verified by functional magnetic resonance imaging (fMRI) that a large region in occipitotemporal cortex processes the shape of both meaningful and unfamiliar objects (Op de Beeck et al., 2008). In our method, a connection will be built between each unfamiliar novel class in $\mathcal{C}^{\text{novel}}$ and each related well-perceived familiar class in $\mathcal{C}^{\text{base}}$. So the first step is to identify the most relevant base classes for a specific task $\mathcal{T}$. Concretely:

**Definition 3.4** (Surrogate Classes). In episode $\mathcal{T}$, given the set of prototypes $\{\boldsymbol{c}_k\}_{k=1}^K$ for the support set $\mathcal{S}$ and the set of prototypes $\{\boldsymbol{c}_t'\}_{t=1}^{|\mathcal{C}^{\text{base}}|}$ for the base set $\mathcal{D}$, the surrogate classes for episode classes $\mathcal{C}^{\mathcal{T}}$ is given as:

$$\mathcal{C}^{\text{surrogate}}(\mathcal{T}) = \bigcup_{k=1}^K \underset{t \in \{1, ..., |\mathcal{C}^{\text{base}}|\}}{\text{Top-}R} d(\boldsymbol{c}_k, \boldsymbol{c}_t') \tag{6}$$

in which the top-$R$ function returns $R$ base class indices with smallest distances to $\boldsymbol{c}_k$, and the center for a base class $t$ is given as $\boldsymbol{c}_t' = \frac{1}{|\{(\boldsymbol{x}, y) | \boldsymbol{x} \in \mathcal{D}, y=t\}|} \sum_{\boldsymbol{x} \in \mathcal{D}, y=t} \phi(\boldsymbol{x})$. Here $R$ is a hyperparameter.

The rationale behind this selection instead of simply using the whole base classes $\mathcal{C}^{\text{base}}$ is that, the episode classes $\mathcal{C}^{\mathcal{T}}$ are only overlapped with a portion of base classes (Zhang et al., 2021a), and

discriminative similarities are likely to be overwhelmed by the background signal especially when the number of base classes is large. After the surrogate classes are found, we re-index their feature centers to be $\{c'_j\}_{j=1}^{\tilde{R}}, \tilde{R} \leq R \cdot K$. Then, both support centers $\{c_k\}_{k=1}^{K}$ and query feature $z$ are represented by the collection of similarities to these surrogate centers:

$$
\begin{aligned}
d'_k &= (d(c_k, c'_1), ..., d(c_k, c'_{\tilde{R}})), k = 1, ..., K \\
d' &= (d(z, c'_1), ..., d(z, c'_{\tilde{R}}))
\end{aligned}
\tag{7}
$$

where $d'_k, d' \in \mathbb{R}^{\tilde{R}}$ are the surrogate representation for novel class $k$ and query feature $z$, respectively. By surrogate representation, the prediction is found through $\hat{y} = \arg\min_k d(d', d'_k) = \arg\min_k ||d' - d'_k||_2^2$. This set of discriminative distances is rewritten as $d'' \in \mathbb{R}^K$ such that $d''_k = d(d', d'_k)$. An illustration of the surrogate representation is shown in Figure 1 on Multi-DigitMNIST, a demonstrative dataset.

**Integrating Feature Representation and Surrogate Representation.** Until now, we have two discriminative systems, i.e., feature-based $d \in \mathbb{R}^K$ and surrogate-based $d'' \in \mathbb{R}^K$. A natural idea is to combine them to form the following final criterion:

$$
\tilde{d} = \beta \frac{d}{||d||_1} + \gamma \frac{d''}{||d''||_1},
\tag{8}
$$

where $d$ and $d''$ are normalized by their Manhattan norm, $||d||_1 = \sum_{k=1}^{K} d_k$ and $||d''||_1 = \sum_{k=1}^{K} d''_k$, respectively, and $\beta$ and $\gamma$ are two hyperparameters adjusting the weights for feature representation and surrogate representation.

## 3.4 DEEPVORO: INTEGRATING MULTI-LEVEL HETEROGENEITY OF FSL

In this section we present DeepVoro, a fast geometric ensemble framework that unites our contributions to multiple stages of FSL, and show how it can be promoted to DeepVoro++ by incorporating surrogate representation.

**Compositional Feature Transformation.** It is believed that FSL algorithms favor features with more Gaussian-like distributions, and thus various kinds of transformations are used to improve the normality of feature distribution, including power transformation (Hu et al., 2021), Tukey's Ladder of Powers Transformation (Yang et al., 2021), and L2 normalization (Wang et al., 2019). While these transformations are normally used independently, here we propose to combine several transformations sequentially in order to enlarge the expressivity of transformation function and to increase the polymorphism of the FSL process. Specifically, for a feature vector $z$, three kinds of transformations are considered: (I) *L2 Normalization.* By projection onto the unit sphere in $\mathbb{R}^n$, the feature is normalized as: $f(z) = \frac{z}{||z||_2}$. (II) *Linear Transformation.* Now since all the features are located on the unit sphere, we then can do scaling and shifting via a linear transformation: $g_{w,b}(z) = wz + b$. (III) *Tukey's Ladder of Powers Transformation.* Finally, Tukey's Ladder of Powers Transformation is applied on the feature: $h_\lambda(z) = \begin{cases} z^\lambda & \text{if } \lambda \neq 0 \\ \log(z) & \text{if } \lambda = 0 \end{cases}$. By the composition of L2 normalization, linear transformation, and Tukey's Ladder of Powers Transformation, now the transformation function becomes $(h_\lambda \circ g_{w,b} \circ f)(z)$ parameterized by $w, b, \lambda$.

**Multi-level Heterogeneities in FSL.** Now we are ready to articulate the hierarchical heterogeneity existing in different stages of FSL. (I) *Feature-level Heterogeneity*: Data augmentation has been exhaustively explored for expanding the data size of FSL (Ni et al., 2021), including but not limited to rotation, flipping, cropping, erasing, solarization, color jitter, MixUp (Zhang et al., 2017), etc. The modification of image $x$ will change the position of feature $z$ in the feature space. We denote all possible translations of image as a set of functions $\{T\}$. (II) *Transformation-level Heterogeneity*: After obtaining the feature $z$, a parameterized transformation is applied to it, and the resulting features can be quite different for these parameters (see Figure F.1). We denote the set of all possible transformations to be $\{P_{w,b,\lambda}\}$. (III) *Geometry-level Heterogeneity*: Even with the provided feature, the few-shot classification model can still be diverse: whether a VD or PD-based model is used, whether the feature or the surrogate representation is adopted, and the setting of $R$ will also change the degree of locality. We denote all possible models as $\{M\}$.

**DeepVoro for Fast Geometric Ensemble of VDs.** With the above three-layer heterogeneity, the FSL process can be encapsulated as $(M \circ P_{w,b,\lambda} \circ \phi \circ T)(\boldsymbol{x})$, and all possible configurations of a given episode $\mathcal{T}$ with a fixed $\phi$ is the Cartesian product of these three sets: $\{T\} \times \{P_{w,b,\lambda}\} \times \{M\}$. Indeed, when a hold-out validation dataset is available, it can be used to find the optimal combination, but by virtue of ensemble learning, multiple models can still contribute positively to FSL (Dvornik et al., 2019). Since the cardinality of the resulting configuration set could be very large, the FSL model $M$ as well as the ensemble algorithm is required to be highly efficient. The VD is a nonparametric model and no training is needed during the meta-testing stage, making it suitable for fast geometric ensemble. While CIVD models the cluster-to-point relationship via an influence function, here we further extend it so that cluster-to-cluster relationship can be considered. This motivates us to define Cluster-to-cluster Voronoi Diagram (CCVD):

**Definition 3.5** (Cluster-to-cluster Voronoi Diagram). Let $\Omega = \{\omega_1, ..., \omega_K\}$ be a partition of the space $\mathbb{R}^n$, and $\mathcal{C} = \{\mathcal{C}_1, ..., \mathcal{C}_K\}$ be a set of *totally ordered sets* with the same cardinality $L$ (i.e. $|\mathcal{C}_1| = |\mathcal{C}_2| = ... = |\mathcal{C}_K| = L$). The set of pairs $\{(\omega_1, \mathcal{C}_1), ..., (\omega_K, \mathcal{C}_K)\}$ is a Cluster-to-cluster Voronoi Diagram (CCVD) with respect to the influence function $F(\mathcal{C}_k, \mathcal{C}(\boldsymbol{z}))$, and each cell is obtained via $\omega_r = \{\boldsymbol{z} \in \mathbb{R}^n : r(\boldsymbol{z}) = r\}, r \in \{1, .., K\}$, with

$$r(\boldsymbol{z}) = \underset{k \in \{1, ..., K\}}{\arg \max} F(\mathcal{C}_k, \mathcal{C}(\boldsymbol{z})) \tag{9}$$

where $\mathcal{C}(\boldsymbol{z})$ is the cluster (also a totally ordered set with cardinality $L$) that query point $\boldsymbol{z}$ belongs, which is to say, all points in this cluster (query cluster) will be assigned to the same cell. Similarly, the Influence Function is defined upon two totally ordered sets $\mathcal{C}_k = \{\boldsymbol{c}_k^{(i)}\}_{i=1}^L$ and $\mathcal{C}(\boldsymbol{z}) = \{\boldsymbol{z}^{(i)}\}_{i=1}^L$:

$$F(\mathcal{C}_k, \mathcal{C}(\boldsymbol{z})) = -\operatorname{sign}(\alpha) \sum_{i=0}^L d(\boldsymbol{c}_k^{(i)}, \boldsymbol{z}^{(i)})^\alpha. \tag{10}$$

With this definition, now we are able to streamline our aforementioned novel approaches into a single ensemble model. Suppose there are totally $L$ possible settings in our configuration pool $\{T\} \times \{P_{w,b,\lambda}\} \times \{M\}$, for all configurations $\{\rho_i\}_{i=1}^L$, we apply them onto the support set $\mathcal{S}$ to generate the $K$ totally ordered clusters $\{\{\boldsymbol{c}_k^{(\rho_i)}\}_{i=1}^L\}_{k=1}^K$ including each center $\boldsymbol{c}_k^{(\rho_i)}$ derived through configuration $\rho_i$, and onto a query sample $\boldsymbol{x}$ to generate the query cluster $\mathcal{C}(\boldsymbol{z}) = \{\boldsymbol{z}^{(\rho_1)}, ..., \boldsymbol{z}^{(\rho_L)}\}$, and then plug these two into Definition 3.5 to construct the final Voronoi Diagram.

When only the feature representation is considered in the configuration pool, i.e. $\rho_i \in \{T\} \times \{P_{w,b,\lambda}\}$, our FSL process is named as DeepVoro; if surrogate representation is also incorporated, i.e. $\rho_i \in \{T\} \times \{P_{w,b,\lambda}\} \times \{M\}$, DeepVoro is promoted to DeepVoro++ that allows for higher geometric diversity. See Appendix A for a summary of the notations and acronyms

## 4 EXPERIMENTS

The main goals of our experiments are to: (1) validate the strength of CIVD to integrate parametric and nonparametric classifiers and confirm the necessity of Voronoi reduction; (2) investigate how different levels of heterogeneity individually or collaboratively contribute to the overall result, and compare them with the state-of-art method; (3) reanalyze this ensemble when the surrogate representation comes into play, and see how it could ameliorate the extreme shortage of support samples. See Table 2 for a summary and Appendix D for the detailed descriptions of mini-ImageNet (Vinyals et al., 2016), CUB (Welinder et al., 2010), and tiered-ImageNet (Ren et al., 2018), that are used in this paper.

Table 2: Summarization of the datasets used in the paper.

| Datasets | Base classes | Novel classes | Image size | Images per class |
|---|---|---|---|---|
| MultiDigitMNIST | 64 | 20 | $64 \times 64 \times 1$ | 1000 |
| mini-ImageNet | 64 | 20 | $84 \times 84 \times 3$ | 600 |
| CUB | 100 | 50 | $84 \times 84 \times 3$ | $44 \sim 60$ |
| tiered-ImageNet | 351 | 160 | $84 \times 84 \times 3$ | $732 \sim 1300$ |

**DeepVoro$--$: Integrating Parametric and Nonparametric Methods via CIVD.** To verify our proposed CIVD model for the integration of parameter/nonparametric FSL classifiers, we first run three standalone models: Logistic Regressions with Power/Voronoi Diagrams as the underlining geometric structure (Power-LR/Voronoi-LR), and vanilla Voronoi Diagram (VD, i.e. nearest neighbor model), and then integrate VD with either Power/Voronoi-LR (see Appendix E for details). Interestingly, VD with the Power-LR has never reached the best result, suggesting that ordinary LR cannot

be integrated with VD due to their intrinsic distinct geometric structures. After the proposed Voronoi reduction (Theorem 3.1), however, VD+Voronoi-LR is able to improve upon both models in most cases, suggesting that CIVD can ideally integrate parameter and nonparametric models for better FSL.

Table 3: The 5-way few-shot accuracy (in %) with 95% confidence intervals of DeepVoro and DeepVoro++ compared against the state-of-the-art results on three benchmark datasets. ¶ The results of DC and S2M2_R are reproduced based on open-sourced implementations using the same random seed with DeepVoro.

| Methods | mini-ImageNet | | CUB | | tiered-ImageNet | |
|---|---|---|---|---|---|---|
| | 5way 1shot | 5way 5shot | 5way 1shot | 5way 5shot | 5way 1shot | 5way 5shot |
| MAML (Finn et al., 2017) | $54.69 \pm 0.89$ | $66.62 \pm 0.83$ | $71.29 \pm 0.95$ | $80.33 \pm 0.70$ | $51.67 \pm 1.81$ | $70.30 \pm 0.08$ |
| Meta-SGD (Li et al., 2017) | $50.47 \pm 1.87$ | $64.03 \pm 0.94$ | $53.34 \pm 0.97$ | $67.59 \pm 0.82$ | – | – |
| Meta Variance Transfer (Park et al., 2020) | – | $67.67 \pm 0.70$ | – | $80.33 \pm 0.61$ | – | – |
| MetaGAN (Zhang et al., 2018) | $52.71 \pm 0.64$ | $68.63 \pm 0.67$ | – | – | – | – |
| Delta-Encoder (Schwartz et al., 2018) | 59.9 | 69.7 | 69.8 | 82.6 | – | – |
| Matching Net (Vinyals et al., 2016) | $64.03 \pm 0.20$ | $76.32 \pm 0.16$ | $73.49 \pm 0.89$ | $84.45 \pm 0.58$ | $68.50 \pm 0.92$ | $80.60 \pm 0.71$ |
| Prototypical Net (Snell et al., 2017) | $54.16 \pm 0.82$ | $73.68 \pm 0.65$ | $72.99 \pm 0.88$ | $86.64 \pm 0.51$ | $65.65 \pm 0.92$ | $83.40 \pm 0.65$ |
| Baseline++ (Chen et al., 2019a) | $57.53 \pm 0.10$ | $72.99 \pm 0.43$ | $70.40 \pm 0.81$ | $82.92 \pm 0.78$ | $60.98 \pm 0.21$ | $75.93 \pm 0.17$ |
| Variational Few-shot (Zhang et al., 2019) | $61.23 \pm 0.26$ | $77.69 \pm 0.17$ | – | – | – | – |
| TriNet (Chen et al., 2019b) | $58.12 \pm 1.37$ | $76.92 \pm 0.69$ | $69.61 \pm 0.46$ | $84.10 \pm 0.35$ | – | – |
| LEO (Rusu et al., 2018) | $61.76 \pm 0.08$ | $77.59 \pm 0.12$ | $68.22 \pm 0.22$ | $78.27 \pm 0.16$ | $66.33 \pm 0.05$ | $81.44 \pm 0.09$ |
| DCO (Lee et al., 2019) | $62.64 \pm 0.61$ | $78.63 \pm 0.46$ | – | – | $65.99 \pm 0.72$ | $81.56 \pm 0.53$ |
| Negative-Cosine (Liu et al., 2020a) | $63.85 \pm 0.81$ | $81.57 \pm 0.56$ | $72.66 \pm 0.85$ | $89.40 \pm 0.43$ | – | – |
| MTL (Wang et al., 2021) | $59.84 \pm 0.22$ | $77.72 \pm 0.09$ | – | – | $67.11 \pm 0.12$ | $83.69 \pm 0.02$ |
| ConstellationNet (Xu et al., 2021) | $64.89 \pm 0.23$ | $79.95 \pm 0.17$ | – | – | – | – |
| AFHN (Li et al., 2020b) | $62.38 \pm 0.72$ | $78.16 \pm 0.56$ | $70.53 \pm 1.01$ | $83.95 \pm 0.63$ | – | – |
| AM3+TRAML (Li et al., 2020a) | $67.10 \pm 0.52$ | $79.54 \pm 0.60$ | – | – | – | – |
| E3BM (Liu et al., 2020c) | $63.80 \pm 0.40$ | $80.29 \pm 0.25$ | – | – | $71.20 \pm 0.40$ | $85.30 \pm 0.30$ |
| SimpleShot (Wang et al., 2019) | $64.29 \pm 0.20$ | $81.50 \pm 0.14$ | – | – | $71.32 \pm 0.22$ | $86.66 \pm 0.15$ |
| R2-D2 (Liu et al., 2020b) | $65.95 \pm 0.45$ | $81.96 \pm 0.32$ | – | – | – | – |
| Robust-dist++ (Dvornik et al., 2019) | $63.73 \pm 0.62$ | $81.19 \pm 0.43$ | – | – | $70.44 \pm 0.32$ | $85.43 \pm 0.21$ |
| IEPT (Zhang et al., 2021b) | $67.05 \pm 0.44$ | $82.90 \pm 0.30$ | – | – | $72.24 \pm 0.50$ | $86.73 \pm 0.34$ |
| MELR (Fei et al., 2021) | $67.40 \pm 0.43$ | $83.40 \pm 0.28$ | $70.26 \pm 0.50$ | $85.01 \pm 0.32$ | $72.14 \pm 0.51$ | $87.01 \pm 0.35$ |
| S2M2_R¶ (Mangla et al., 2020) | $64.65 \pm 0.45$ | $83.20 \pm 0.30$ | $80.14 \pm 0.45$ | $90.99 \pm 0.23$ | $68.12 \pm 0.52$ | $86.71 \pm 0.34$ |
| M-SVM+MM+ens+val (Ni et al., 2021) | $67.37 \pm 0.32$ | $84.57 \pm 0.21$ | – | – | – | – |
| DeepEMD (Zhang et al., 2020a) | $65.91 \pm 0.82$ | $82.41 \pm 0.56$ | $75.65 \pm 0.83$ | $88.69 \pm 0.50$ | $71.16 \pm 0.87$ | $86.03 \pm 0.58$ |
| DeepEMD-V2 (Zhang et al., 2020b) | $68.77 \pm 0.29$ | $84.13 \pm 0.53$ | $79.27 \pm 0.29$ | $89.80 \pm 0.51$ | $74.29 \pm 0.32$ | $86.98 \pm 0.60$ |
| DC¶ (Yang et al., 2021) | $67.79 \pm 0.45$ | $83.69 \pm 0.31$ | $79.93 \pm 0.46$ | $90.77 \pm 0.24$ | $74.24 \pm 0.50$ | $88.38 \pm 0.31$ |
| PT+NCM (Hu et al., 2021) | $65.35 \pm 0.20$ | $83.87 \pm 0.13$ | $80.57 \pm 0.20$ | $91.15 \pm 0.10$ | $69.96 \pm 0.22$ | $86.45 \pm 0.15$ |
| **DeepVoro** | $69.48 \pm 0.45$ | $\mathbf{86.75 \pm 0.28}$ | $\mathbf{82.99 \pm 0.43}$ | $\mathbf{92.62 \pm 0.22}$ | $74.98 \pm 0.48$ | $\mathbf{89.40 \pm 0.29}$ |
| **DeepVoro++** | $\mathbf{71.30 \pm 0.46}$ | $85.40 \pm 0.30$ | $82.95 \pm 0.43$ | $91.21 \pm 0.23$ | $\mathbf{75.38 \pm 0.48}$ | $87.25 \pm 0.33$ |

**DeepVoro: Improving FSL by Hierarchical Heterogeneities.** In this section, we only consider two levels of heterogeneities for ensemble: feature-level and transformation-level. For feature-level ensemble, we utilize three kinds of image augmentations: rotation, flipping, and central cropping summing up to 64 distinct ways of data augmentation (Appendix F). For transformation-level ensemble, we use the proposed compositional transformations with 8 different combinations of $\lambda$ and $b$ that encourage a diverse feature transformations (Appendix F.1) without loss of accuracy (Figure 2). The size of the resulting configuration pool becomes 512 and DeepVoro's performance is shown in Table 3. Clearly, DeepVoro outperforms all previous methods especially on 5-way 5-shot FSL. Specifically, DeepVoro is better than the next best by 2.18% (than Ni et al. (2021)) on mini-ImageNet, by 1.47% (than Hu et al. (2021)) on CUB, and by 1.02% (than Yang et al. (2021)) on tiered-ImageNet. Note that this is an estimated improvement because not all competitive methods here are tested with the same random seed and the number of episodes. More detailed results can be found in Appendix F. By virtue of CCVD and using the simplest VD as the building block, DeepVoro is arguably able to yield a consistently better result by the ensemble of a massive pool of independent VD. DeepVoro also exhibits high resistance to outliers, as shown in Figure K.16.

**DeepVoro++: Further Improvement of FSL via Surrogate Representation.** In surrogate representation, the number of neighbors $R$ for each novel class and the weight $\beta$ balancing surrogate/feature representations are two hyperparameters. With the help of an available validation set, a natural question is that whether the hyperparameter can be found through the optimization on the validation set, which requires a good generalization of the hyperparameters across different novel classes. From Figure K.13, the accuracy of VD with varying hyperparameter shows a good agreement between testing and validation sets. With this in mind, we select 10 combinations of $\beta$ and $R$, guided by the validation set, in conjunction with 2 different feature transformations and 64 different image augmentations, adding up to a large pool of 1280 configurations for ensemble (denoted by DeepVoro++). As shown in Table 3, DeepVoro++ achieves best results for 1-shot FSL — 2.53%

Table 4: DeepVoro ablation experiments with feature(Feat.)/transformation(Trans.)/geometry(Geo.)-level heterogeneities on mini-ImageNet 5-way few-shot dataset. $L$ denotes the size of configuration pool, i.e. the number of ensemble members. ♯These lines show the average VD accuracy without CCVD integration.

| Methods | Geometric Structures | Feat. | Trans. | Geo. | $L$ | 5-way 1-shot | 5-way 5-shot |
|---|---|---|---|---|---|---|---|
| | *tunable parameters:* | rotation etc. | $w, b, \lambda$ | $\beta, \gamma, R$ | | | |
| DeepVoro−− | CIVD | ✗ | ✗ | ✗ | − | $65.85 \pm 0.43$ | $84.66 \pm 0.29$ |
| DeepVoro | CCVD | ✗ | ✗ | ✗ | 1♯ | $66.92 \pm 0.45$ | $84.64 \pm 0.30$ |
| | | ✗ | 8 | ✗ | 8 | $66.45 \pm 0.44$ | $84.55 \pm 0.29$ |
| | | 64 | ✗ | ✗ | 64 | $67.88 \pm 0.45$ | $86.39 \pm 0.29$ |
| | | 64 | 8 | ✗ | 512 | $69.48 \pm 0.45$ | $86.75 \pm 0.28$ |
| DeepVoro++ | CCVD w/ surrogate representation | ✗ | ✗ | ✗ | 1♯ | $68.68 \pm 0.46$ | $84.28 \pm 0.31$ |
| | | ✗ | 2 | 10 | 20 | $68.38 \pm 0.46$ | $83.27 \pm 0.31$ |
| | | 64 | ✗ | ✗ | 64 | $70.95 \pm 0.46$ | $84.77 \pm 0.30$ |
| | | 64 | 2 | 10 | 1280 | $71.30 \pm 0.46$ | $85.40 \pm 0.30$ |

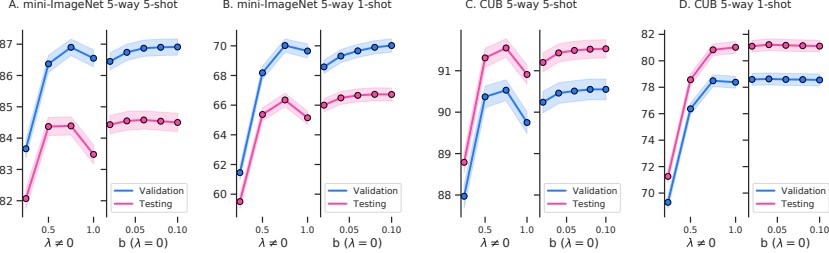

Figure 2: The 5-way few-shot accuracy of VD with different $\lambda$ and $b$ on mini-ImageNet and CUB Datasets.

higher than Zhang et al. (2020b), $2.38\%$ higher than Hu et al. (2021), and $1.09\%$ higher than Zhang et al. (2020b), on three datasets, respectively, justifying the efficacy of our surrogate representation. See Appendix G for more detailed analysis.

**Ablation Experiments and Running Time.** Table 4 varies the level of heterogeneity (see Table F.4 and G.5 for all datasets). The average accuracy of VDs without CCVD integration is marked by ♯, and is significantly lower than the fully-fledged ensemble. Table 5 presents the running time of DeepVoro(++) benchmarked in a 20-core Intel© Core$^{\text{TM}}$ i7 CPU with NumPy (v1.20.3), whose efficiency is comparable to DC/S2M2_2, even with $>1000\times$ diversity.

**Experiments with Different Backbones, Meta-training Protocols, and Domains.** Because different feature extraction backbones, meta-training losses, and degree of discrepancy between the source/target domains will all affect the downstream FSL, we here examine the robustness of DeepVoro/DeepVoro++ under a number of different circumstances, and details are shown in Appendices H, I, J. Notably, DeepVoro/DeepVoro++ attains the best performance by up to $5.55\%$, and is therefore corroborated as a superior method for FSL, regardless of the backbone, training loss, or domain.

Table 5: Running time comparison.

| Methods | | Time (min) |
|---|---|---|
| DC | | 88.29 |
| S2M2_R | | 33.89 |
| #ensemble members: | | |
| DeepVoro | 1 | 0.05 |
| | 512 | 25.67 |
| DeepVoro++ | 1 | 0.14 |
| | 1280 | 179.05 |

## 5 CONCLUSION

In this paper, our contribution is threefold. We first theoretically unify parametric and nonparametric few-shot classifiers into a general geometric framework (VD) and show an improved result by virtue of this integration (CIVD). By extending CIVD to CCVD, we present a fast geometric ensemble method (DeepVoro) that takes into consideration thousands of FSL configurations with high efficiency. To deal with the extreme data insufficiency in one-shot learning, we further propose a novel surrogate representation which, when incorporated into DeepVoro, promotes the performance of one-shot learning to a higher level (DeepVoro++). In future studies, we plan to extend our geometric approach to meta-learning-based FSL and lifelong FSL.

ACKNOWLEDGMENTS

This research was supported in part by NSF through grant IIS-1910492.

REPRODUCIBILITY STATEMENT

Our code as well as data split, random seeds, hyperparameters, scripts for reproducing the results in the paper are available at https://github.com/horsepurve/DeepVoro.

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

# A  NOTATIONS AND ACRONYMS

Table A.1: Notations and acronyms for VD, PD, CIVD, and CCVD, four geometric structures discussed in the paper.

| Geometric Structures | Acronyms | Notations | Description |
|---|---|---|---|
| Voronoi Diagram | VD | $\boldsymbol{c}_k$ | center for a Voronoi cell $\omega_k, k \in \{1, .., K\}$ |
| | | $\omega_k$ | dominating region for center $\boldsymbol{c}_k, k \in \{1, .., K\}$ |
| Power Diagram | PD | c | center for a Power cell $\omega_k, k \in \{1, .., K\}$ |
| | | $\nu_k$ | weight for center $\boldsymbol{c}_k, k \in \{1, .., K\}$ |
| | | $\omega_k$ | dominating region for center $\boldsymbol{c}_k, k \in \{1, .., K\}$ |
| Cluster-induced Voronoi Diagram | CIVD | $\mathcal{C}_k$ | cluster as the "center" for a CIVD cell $\omega_k, k \in \{1, .., K\}$ |
| | | $\omega_k$ | dominating region for cluster $\mathcal{C}_k$ |
| | | $F$ | influence function $F(\mathcal{C}_k, \boldsymbol{z})$ from cluster $\mathcal{C}_k$ to query point $\boldsymbol{z}$ |
| | | $\alpha$ | magnitude of the influence |
| Cluster-to-cluster Voronoi Diagram | CCVD | $\mathcal{C}_k$ | cluster as the "center" for a CCVD cell $\omega_k, k \in \{1, .., K\}$ |
| | | $\omega_k$ | dominating region for cluster $\mathcal{C}_k$ |
| | | $\mathcal{C}(\boldsymbol{z})$ | the cluster that query point $\boldsymbol{z}$ belongs |
| | | $F$ | influence function $F(\mathcal{C}_k, \mathcal{C}(\boldsymbol{z}))$ from $\mathcal{C}_k$ to query cluster $\mathcal{C}(\boldsymbol{z})$ |
| | | $\alpha$ | magnitude of the influence |

Table A.2: Summary and comparison of geometric structures, centers, tunable parameters, and the numbers of tunable parameters (denoted by #) for DeepVoro−−, DeepVoro, and DeepVoro++. Parameters for feature-level , transformation-level , and geometry-level heterogeneity are shown in yellow , blue , and red , respectively. See Sec. F for implementation details. †Here PD is reduced to VD by Theorem 3.1. ‡For every $\lambda$ (or $R$), the $b$ (or $\beta$) value with the highest validation accuracy is introduced into the configuration pool.

| Methods | Geometric Structures | Centers | Tunable Param. | # | Description |
|---|---|---|---|---|---|
| **DeepVoro−−** | CIVD | $\mathcal{C}_k = \{\boldsymbol{c}_k, \tilde{\boldsymbol{c}}_k\}$ $\boldsymbol{c}_k$ from VD $\tilde{\boldsymbol{c}}_k$ from PD† | − | − | − |
| **DeepVoro** | CCVD | $\mathcal{C}_k = \{\boldsymbol{c}_k^{(\rho_i)}\}_{i=1}^L$ $\rho_i \in \{T\} \times \{P_{w,b,\lambda}\}$ | angle of rotation | 4 | − |
| | | | flipping or not | 2 | − |
| | | | scaling & cropping | 8 | − |
| | | | $w = 1$ | − | scale factor in linear transformation |
| | | | $b$ | 4 | shift factor in linear transformation |
| | | | $\lambda$ | 2 | exponent in powers transformation |
| | | | #configurations $L = 512$ | | |
| **DeepVoro++** | CCVD | $\mathcal{C}_k = \{\boldsymbol{c}_k^{(\rho_i)}\}_{i=1}^L$ $\rho_i \in \{T\} \times \{P_{w,b,\lambda}\} \times \{M\}$ | angle of rotation | 4 | − |
| | | | flipping or not | 2 | − |
| | | | scaling & cropping | 8 | − |
| | | | $w = 1$ | − | scale factor in linear transformation |
| | | | $b$ | 1‡ | shift factor in linear transformation |
| | | | $\lambda$ | 2 | exponent in powers transformation |
| | | | $R$ | 10 | the number of top-R nearest base prototypes for a novel prototype |
| | | | $\gamma = 1$ | − | weight for surrogate representation |
| | | | $\beta$ | 1‡ | weight for feature representation |
| | | | #configurations $L = 1280$ | | |

# B  POWER DIAGRAM SUBDIVISION AND VORONOI REDUCTION

## B.1  PROOF OF THEOREM 3.1

**Lemma B.1.** The vertical projection from the lower envelope of the hyperplanes $\{\Pi_k(\boldsymbol{z}) : \boldsymbol{W}_k^T \boldsymbol{z} + b_k\}_{k=1}^K$ onto the input space $\mathbb{R}^n$ defines the cells of a PD.

**Theorem 3.1** (Voronoi Diagram Reduction). The linear classifier parameterized by $\boldsymbol{W}, \boldsymbol{b}$ partitions the input space $\mathbb{R}^n$ to a Voronoi Diagram with centers $\{\tilde{\boldsymbol{c}}_1, ..., \tilde{\boldsymbol{c}}_K\}$ given by $\tilde{\boldsymbol{c}}_k = \frac{1}{2}\boldsymbol{W}_k$ if $b_k = -\frac{1}{4}\|\boldsymbol{W}_k\|_2^2, k = 1, ..., K$.

*Proof.* We first articulate Lemma B.1 and find the exact relationship between the hyperplane $\Pi_k(\boldsymbol{z})$ and the center of its associated cell in $\mathbb{R}^n$. By Definition 3.1, the cell for a point $\boldsymbol{z} \in \mathbb{R}^n$ is found by comparing $d(\boldsymbol{z}, \boldsymbol{c}_k)^2 - \nu_k$ for different $k$, so we define the power function $p(\boldsymbol{z}, S)$ expressing this value

$$p(\boldsymbol{z}, S) = (\boldsymbol{z} - \boldsymbol{u})^2 - r^2 \tag{11}$$

in which $S \subseteq \mathbb{R}^n$ is a sphere with center $\boldsymbol{u}$ and radius $r$. In fact, the weight $\nu$ associated with a center in Definition 3.1 can be intepreted as the square of the radius $r^2$. Next, let $U$ denote a paraboloid $y = \boldsymbol{z}^2$, let $\Pi(S)$ be the transform that maps sphere $S$ with center $\boldsymbol{u}$ and radius $r$ into hyperplane

$$\Pi(S) : y = 2\boldsymbol{z} \cdot \boldsymbol{u} - \boldsymbol{u} \cdot \boldsymbol{u} + r^2. \tag{12}$$

It can be proved that $\Pi$ is a bijective mapping between arbitrary spheres in $\mathbb{R}^n$ and nonvertical hyperplanes in $\mathbb{R}^{n+1}$ that intersect $U$ (Aurenhammer, 1987). Further, let $\boldsymbol{z}'$ denote the vertical projection of $\boldsymbol{z}$ onto $U$ and $\boldsymbol{z}''$ denote its vertical projection onto $\Pi(S)$, then the power function can be written as

$$p(\boldsymbol{z}, S) = d(\boldsymbol{z}, \boldsymbol{z}') - d(\boldsymbol{z}, \boldsymbol{z}''), \tag{13}$$

which implies the following relationships between a sphere in $\mathbb{R}^n$ and an associated hyperplane in $\mathbb{R}^{n+1}$ (Lemma 4 in Aurenhammer (1987)): let $S_1$ and $S_2$ be nonco-centeric spheres in $\mathbb{R}^n$, then the bisector of their Power cells is the vertical projection of $\Pi(S_1) \cap \Pi(S_2)$ onto $\mathbb{R}^n$. Now, we have a direct relationship between sphere $S$, and hyperplane $\Pi(S)$, and comparing equation (12) with the hyperplanes used in logistic regression $\{\Pi_k(\boldsymbol{z}) : \boldsymbol{W}_k^T \boldsymbol{z} + b_k\}_{k=1}^K$ gives us

$$\begin{aligned} \boldsymbol{u} &= \frac{1}{2}\boldsymbol{W}_k \\ r^2 &= b_k + \frac{1}{4}||\boldsymbol{W}_k||_2^2. \end{aligned} \tag{14}$$

Although there is no guarantee that $b_k + \frac{1}{4}||\boldsymbol{W}_k||_2^2$ is always positive for an arbitrary logistic regression model, we can impose a constraint on $r^2$ to keep it be zero during the optimization, which implies

$$b_k = -\frac{1}{4}||\boldsymbol{W}_k||_2^2. \tag{15}$$

By this way, the radii for all $K$ spheres become identical (all zero). After the optimization of logistic regression model, the centers $\{\frac{1}{2}\boldsymbol{W}_k\}_{k=1}^K$ will be used for CIVD integration. $\square$

## C    DETAILS ABOUT THE DEMONSTRATIVE EXAMPLE ON MULTIDIGITMNIST DATASET

MultiDigitMNIST (Sun, 2019) dataset is created by concatenating two (or three) digits of different classes from MNIST for few-shot image classification. Here we use DoubleMNIST Datasets (i.e. two digits in an image) consisting of 100 classes (00 to 09), 1000 images of size $64 \times 64 \times 1$ per class, and the classes are further split into 64, 20, and 16 classes for training, testing, and validation, respectively. To better embed into the $\mathbb{R}^2$ space, we pick a ten-classes subset (00, 01, 12, 13, 04, 05, 06, 77, 08, and 09) as the base classes for meta-training, and another five-class subset (02, 49, 83, 17, and 36) for one episode. The feature extractor is a 4-layer convolutional network with an additional fully-connected layer for 2D embedding. In Figure 1 left panel, the VD is obtained by setting the centroid of each base class as the Voronoi center. For each novel class, the Voronoi center is simply the 1-shot support sample (Figure 1 central panel). The surrogate representation is computed as the collection of distances from a support/query sample to each of the base classes, as shown in Figure 1 right panel. Interestingly, the surrogate representations for a novel class, no matter if it is a support sample (dotted line) or a query sample (colored line) generally follow a certain pattern — akin within a class, distinct cross class — make them ideal surrogates for distinguishing between different novel classes. In our paper, we design a series of algorithms answering multiple questions regarding this surrogate representation: how to select base classes for the calculation of surrogate representation, how to combine it with feature representation, and how to integrate it into the overall ensemble workflow.

## D    MAIN DATASETS

For a fair and thorough comparison with previous works, three widely-adopted benchmark datasets are used throughout this paper.

(1) **mini-ImageNet** (Vinyals et al., 2016) is a shrunk subset of ILSVRC-12 (Russakovsky et al., 2015), consists of 100 classes in which 64 classes for training, 20 classes for testing and 16 classes for validation. Each class has 600 images of size $84 \times 84 \times 3$.

(2) **CUB** (Welinder et al., 2010) is another benchmark dataset for FSL, especially fine-grained FSL, including 200 species (classes) of birds. CUB is an unbalanced dataset with 58 images in average per class, also of size $84 \times 84 \times 3$. We split all classes into 100 base classes, 50 novel classes, and 50 validation classes, following previous works (Chen et al., 2019a).

(3) **tiered-ImageNet** (Ren et al., 2018) is another subset of ILSVRC-12 (Russakovsky et al., 2015) but has more images, 779,165 images in total. All images are categorized into 351 base classes, 97 validation classes, and 160 novel classes. The number of images in each class is not always the same, 1281 in average. The image size is also $84 \times 84 \times 3$.

## E    DEEPVORO−−: INTEGRATING PARAMETRIC AND NONPARAMETRIC METHODS VIA CIVD

Table E.3: Cluster-induced Voronoi Diagram (CIVD) for the integration of parametric Logistic Regression (LR) and nonparametric nearest neighbor (i.e. Voronoi Diagram, VD) methods. The results from S2M2_R and DC are also included in this table but excluded for comparison. Best result is marked in bold.

| Methods | mini-Imagenet | | CUB | | tiered-ImageNet | |
|---|---|---|---|---|---|---|
| | **5-way 1-shot** | **5-way 5-shot** | **5-way 1-shot** | **5-way 5-shot** | **5-way 1-shot** | **5-way 5-shot** |
| S2M2_R | $64.65 \pm 0.45$ | $83.20 \pm 0.30$ | $80.14 \pm 0.45$ | $90.99 \pm 0.23$ | $68.12 \pm 0.52$ | $86.71 \pm 0.34$ |
| DC | $67.79 \pm 0.45$ | $83.69 \pm 0.31$ | $79.93 \pm 0.46$ | $90.77 \pm 0.24$ | $74.24 \pm 0.50$ | $88.38 \pm 0.31$ |
| Power-LR | $65.45 \pm 0.44$ | $84.47 \pm 0.29$ | $\mathbf{79.66 \pm 0.44}$ | $\mathbf{91.62 \pm 0.22}$ | $73.57 \pm 0.48$ | $89.07 \pm 0.29$ |
| Voronoi-LR | $65.58 \pm 0.44$ | $84.51 \pm 0.29$ | $79.63 \pm 0.44$ | $91.61 \pm 0.22$ | $73.65 \pm 0.48$ | $\mathbf{89.15 \pm 0.29}$ |
| VD | $65.37 \pm 0.44$ | $84.37 \pm 0.29$ | $78.57 \pm 0.44$ | $91.31 \pm 0.23$ | $72.83 \pm 0.49$ | $88.58 \pm 0.29$ |
| **CIVD-based DeepVoro−−** | | | | | | |
| VD + Power-LR | $65.63 \pm 0.44$ | $84.25 \pm 0.30$ | $79.52 \pm 0.43$ | $91.52 \pm 0.22$ | $73.68 \pm 0.48$ | $88.71 \pm 0.29$ |
| VD + Voronoi-LR | $\mathbf{65.85 \pm 0.43}$ | $\mathbf{84.66 \pm 0.29}$ | $79.40 \pm 0.44$ | $91.57 \pm 0.22$ | $\mathbf{73.78 \pm 0.48}$ | $89.02 \pm 0.29$ |

### E.1    EXPERIMENTAL SETUP AND IMPLEMENTATION DETAILS

In this section, we first establish three few-shot classification models with different underlying geometric structures, two logistic regression (LR) models and one nearest neighbor model: (1) Power Diagram-based LR (Power-LR), (2) Voronoi Diagram-based LR (Voronoi-LR), and (3) Voronoi Diagram (VD). Then, the main purposes of our analysis are (1) to examine how the performance is affected by the proposed Voronoi Reduction method in Sec. 3.2, and (2) to inspect whether VD can be integrated with Power/Voronoi Diagram-based LRs.

The feature transformation used throughout this section is $P_{w,b,\lambda}$ with $w = 1.0, b = 0.0, \lambda = 0.5$. For Power-LR, we train it directly on the transformed $K$-way $N$-shot support samples using PyTorch library with an Adam optimizer with batch size at 64 and learning rate at 0.01. For Voronoi-LR, the vanilla LR is retrofitted as shown in Algorithm 1, in which the bias is given by Theorem 3.1 to make sure that the parameters induce a VD in each iteration.

In our CIVD model in Definition 3.2, we use a *cluster* instead of a single *prototype* to stand for a novel class. Here this cluster contains two points, i.e. $\mathcal{C}_k = \{\boldsymbol{c}_k, \tilde{\boldsymbol{c}}_k\}$, in which $\boldsymbol{c}_k$ is obtained from VD, and $\tilde{\boldsymbol{c}}_k$ is acquired from Power-LR or Voronoi-LR. The question we intend to answer here is that whether Power-LR or Voronoi-LR is the suitable model for the integration.

---

**Algorithm 1:** Voronoi Diagram-based Logistic Regression.

**Data:** Support Set $\mathcal{S}$
**Result:** $\boldsymbol{W}$

1   Initialize $\boldsymbol{W} \leftarrow \boldsymbol{W}^{(0)}$;
2   **for** *epoch* $\leftarrow 1, ..., \#epoch$ **do**
3      $b_k \leftarrow -\frac{1}{4}||\boldsymbol{W}_k||_2^2, \forall k = 1, ..., K$ ;               ◁ Apply Theorem 3.1
4      Compute loss $\mathcal{L}(\boldsymbol{W}, \boldsymbol{b})$ ;                  ◁ forward propagation
5      Update $\boldsymbol{W}$ ;                          ◁ backward propagation
6   **end**
7   **return** $\boldsymbol{W}$

---

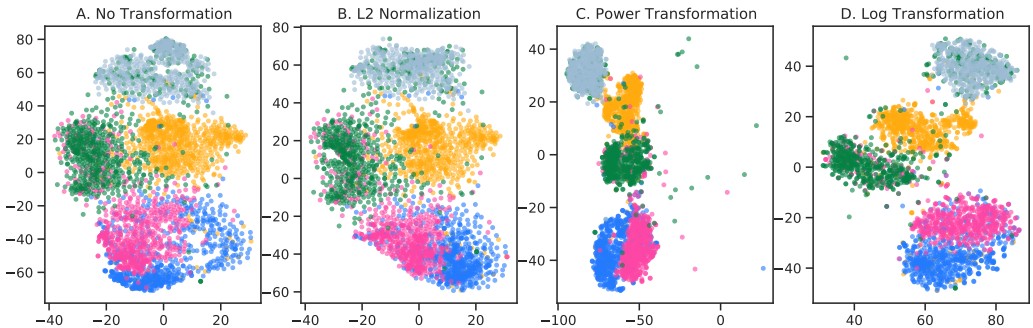

Figure F.1: The t-SNE visualizations of (A) original features, (B) L2 normalization, (C) Tukey's Ladder of Powers Transformation with $\lambda = 0.5$, and (D) compositional transformation with $\lambda = 0, w = 1, b = 0.04$ of 5 novel classes from mini-ImageNet dataset.

### E.2   RESULTS

The results are shown in Table E.3. Interestingly, when integrated with VD, Power-LR never reaches the best result, suggesting that VD and LR are intrinsic different geometric models, and cannot be simply integrated together without additional effort. On mini-ImageNet and tiered-ImageNet datasets, the best results are achieved by either Voronoi-LR or VD+Voronoi-LR, showing that CIVD coupled with the proposed Voronoi reduction can ideally integrate parametric and nonparametric few-shot models. Notably, on these two datasets, when Power-LR is reduced to Voronoi-LR, although the number of parameters is decreased ($\boldsymbol{b}$ is directly given by Theorem 3.1, not involved in the optimization), the performance is always better, for example, increases from $65.45\%$ to $65.58\%$ on 5-way 1-shot mini-ImageNet data. On CUB dataset, results of different models are similar, probably because CUB is a fine-grained dataset and all classes are similar to each other (all birds).

## F   DEEPVORO: IMPROVING FSL VIA HIERARCHICAL HETEROGENEITIES

### F.1   EXPERIMENTAL SETUP AND IMPLEMENTATION DETAILS

In this section we describe feature-level and transformation-level heterogeneities that are used for ensemble in order to improve FSL. See the next section for geometry-level heterogeneity.

**Feature-level heterogeneity.** Considering the reproducibility of the methodology, we only employ deterministic data augmentation upon the images without randomness involved. Specifically, three kinds of data augmentation techniques are used. (1) Rotation is an important augmentation method widely used in self-supervised learning (Mangla et al., 2020). Rotating the original images by $0°$, $90°$, $180°$, and $270°$ gives us four ways of augmentation. (2) After rotation, we can flip the images horizontally, giving rise to additional two choices after each rotation degree. (3) Central cropping after scaling can alter the resolution and focus area of the image. Scaling the original images to $(84+B) \times (84+B)$, $B$ increasing from 0 to 70 with step 10, bringing us eight ways of augmentation.

Finally, different combinations of the three types result in 64 kinds of augmentation methods (i.e. $|\{T\}| = 64$).

**Transformation-level heterogeneity.** In our compositional transformation, the function $(h_\lambda \circ g_{w,b} \circ f)(z)$ is parameterized by $w, b, \lambda$. Since $g$ is appended after the L2 normalization $f$, the vector comes into $g$ is always a unit vector, so we simply set $w = 1$. For the different combinations of $\lambda$ and $b$, we test different values with either $\lambda = 0$ or $\lambda \neq 0$ on the hold-out validation set (as shown in Figure 2 and K.12), and pick top-8 combinations with the best performance on the validation set.

**Ensemble Schemes.** Now, in our configuration pool $\{T\} \times \{P_{w,b,\lambda}\}$, there are 512 possible configurations $\{\rho^{(i)}\}_{i=1}^{512}$. For each $\rho$, we apply it on both the testing and the validation sets. With this large pool of ensemble candidates, how and whether to select a subset $\{\rho^{(i)}\}_{i=1}^{L'} \subseteq \{\rho^{(i)}\}_{i=1}^{512}$ is still a nontrivial problem. Here we explore three different schemes. (1) *Full (vanilla) ensemble*. All candidates in $\{\rho^{(i)}\}_{i=1}^{512}$ are taken into consideration and then plugged into Definition 3.5 to build the CIVD for space partition. (2) *Random ensemble*. A randomly selected subset with size $L' < L$ is used for ensemble. (3) *Guided ensemble*. We expect the performance for $\{\rho^{(i)}\}_{i=1}^{512}$ on the validation set can be used to guide the selection of $\{\rho^{(i)}\}_{i=1}^{L'}$ from the testing set, provided that there is good correlation between the testing set and the validation set. Specifically, we rank the configurations in the validation set with regard to their performance, and add them sequentially into $\{\rho^{(i)}\}_{i=1}^{L'}$ until a maximum ensemble performance is reached on the validation set, then we use this configuration set for the final ensemble. Since VD is nonparametric and fast, we adopt VD as the building block and only use VD for each $\rho$ for the remaining part of the paper. The $\alpha$ value in the influence function (Definition 3.3) is set at 1 throughout the paper, for the simplicity of computation.

For a fair comparison, we downloaded the trained models[1] used by Mangla et al. (2020) and Yang et al. (2021). The performance of FSL algorithms is typically evaluated by a sequence of independent episodes, so the data split and random seed for the selection of novel classes as well as support/query set in each episode will all lead to different result. To ensure the fairness of our evaluation, DC (Yang et al., 2021), and S2M2_R (Mangla et al., 2020) are reevaluated with the same data split and random seed as DeepVoro. The results are obtained by running 2000 episodes and the average accuracy as well as $95\%$ confidence intervals are reported.

## F.2 RESULTS

Table F.4: Ablation study of DeepVoro's performance with different levels of ensemble. The number of ensemble members are given in parentheses.

| Methods | Feature-level | Transformation-level | mini-ImageNet | | CUB | | tiered-ImageNet | |
|---|---|---|---|---|---|---|---|---|
| | | | 1-shot | 5-shot | 1-shot | 5-shot | 1-shot | 5-shot |
| No Ensemble | ✗ | ✗ | $65.37 \pm 0.44$ | $84.37 \pm 0.29$ | $78.57 \pm 0.44$ | $91.31 \pm 0.23$ | $72.83 \pm 0.49$ | $88.58 \pm 0.29$ |
| Vanilla Ensemble (8) | ✗ | ✔ | $66.45 \pm 0.44$ | $84.55 \pm 0.29$ | $80.98 \pm 0.44$ | $91.47 \pm 0.22$ | $74.02 \pm 0.49$ | $88.90 \pm 0.29$ |
| Vanilla Ensemble (64) | ✔ | ✗ | $67.88 \pm 0.45$ | $86.39 \pm 0.29$ | $77.30 \pm 0.43$ | $91.26 \pm 0.23$ | $73.74 \pm 0.49$ | $88.67 \pm 0.29$ |
| Vanilla Ensemble (512) | ✔ | ✔ | $69.23 \pm 0.45$ | $86.70 \pm 0.28$ | $79.90 \pm 0.43$ | $91.70 \pm 0.22$ | $74.51 \pm 0.48$ | $89.11 \pm 0.29$ |
| Random Ensemble (512) | ✔ | ✔ | $69.30 \pm 0.45$ | $86.74 \pm 0.28$ | $80.40 \pm 0.43$ | $91.94 \pm 0.22$ | $74.64 \pm 0.48$ | $89.15 \pm 0.29$ |
| Guided Ensemble (512) | ✔ | ✔ | $\mathbf{69.48 \pm 0.45}$ | $\mathbf{86.75 \pm 0.28}$ | $\mathbf{82.99 \pm 0.43}$ | $\mathbf{92.62 \pm 0.22}$ | $\mathbf{74.98 \pm 0.48}$ | $\mathbf{89.40 \pm 0.29}$ |

Our proposed compositional transformation enlarges the expressivity of the transformation function. When the Tukey's ladder of powers transformation is used individually, as reported in Yang et al. (2021), the optimal $\lambda$ is not 0, but if an additional linear transformation $g$ is inserted between $f$ and $h$, $\lambda = 0$ coupled with a proper $b$ can give even better result, as shown in Figure 2 and K.12. Importantly, from Figure 2, a combination of $\lambda$ and $b$ with good performance on the validation set can also produce satisfactory result on the testing set, suggesting that it is possible to optimize the hyperparameters on the validation set and generalize well on the testing set. In terms of the polymorphism induced by various transformations in the feature space, Figure F.1 exhibits the t-SNE visualizations of the original features and the features after three different kinds of transformations, showing that the relative positions of different novel classes is largely changes especially after compositional transformation (as shown in D). Besides commonly used data augmentation, this transformation provides another level of diversity that may be beneficial to the subsequent ensemble.

The results for different levels of ensemble are shown in Table F.4, in which the number of ensemble member are also indicated. Although transformation ensemble does not involve any change to the feature, it can largely improve the results for 1-shot FSL, from $65.37\%$ to $66.45\%$ on mini-ImageNet,

---

[1]downloaded from `https://github.com/nupurkmr9/S2M2_fewshot`

from $78.57\%$ to $80.98\%$ on CUB, and from $72.83\%$ to $74.02\%$ on tiered-ImageNet, respectively, probably because 1-shot FSL is more prone to overfitting due to its severe data deficiency. Feature-level ensemble, on the other hand, is more important for 5-shot FSL, especially for mini-ImageNet. When combining the two levels together, the number of ensemble members increases to $512$ and the performance significantly surpasses each individual level. On all three datasets, the guided ensemble scheme always achieves the best result for both single-shot and multi-shot cases, showing that the validation set can indeed be used for the guidance of subset selection and our method is robust cross classes in the same domain. When there is no such validation set available, the full ensemble and random ensemble schemes can also give comparable result.

To inspect how performance changes with different number of ensemble members, we exhibit the distribution of accuracy at three ensemble levels for mini-ImageNet in Figure F.2 and F.3 , for CUB in Figure F.4 and F.5, and for tiered-ImageNet in Figure F.6 and F.7. Figure (b) in each of them also exhibits the correlation between the testing and validation sets for all $512$ configurations. Clearly, better result is often reached when there are more configurations for the ensemble, validating the efficacy of our method for improving the performance and robustness for better FSL.

---

**Algorithm 2:** VD with Surrogate Representation for Episode $\mathcal{T}$.

**Data:** Base classes $\mathcal{D}$, Support Set $\mathcal{S} = \{(\boldsymbol{x}_i, y_i)\}_{i=1}^{K \times N}, y_i \in \mathcal{C}^{\mathcal{T}}$, query sample $\boldsymbol{x}$

**Result:** $\tilde{d}$

1   $\mathcal{D}' \leftarrow (P_{w,b,\lambda} \circ \phi \circ T)(\mathcal{D})$ ;          ◁ Extract and transform feature

2   $\mathcal{S}' \leftarrow (P_{w,b,\lambda} \circ \phi \circ T)(\mathcal{S})$;

3   $\boldsymbol{z} \leftarrow (P_{w,b,\lambda} \circ \phi \circ T)(\boldsymbol{x})$;

4   **for** $t \leftarrow 1, ..., |\mathcal{C}^{base}|$;          ◁ Compute prototypes of base classes

5   **do**

6      $\boldsymbol{c}'_t \leftarrow \frac{1}{|\{(\boldsymbol{z}',y)|\boldsymbol{z}' \in \mathcal{D}', y=t\}|} \sum_{\boldsymbol{z}' \in \mathcal{D}', y=t} \boldsymbol{z}'$

7   **end**

8   **for** $k \leftarrow 1, ..., K$;          ◁ Compute prototypes from support samples

9   **do**

10      $\boldsymbol{c}_k \leftarrow \frac{1}{N} \sum_{\boldsymbol{z}' \in \mathcal{S}', y=k} \boldsymbol{z}'$;

11      $d_k \leftarrow d(\boldsymbol{z}, \boldsymbol{c}_k)$

12   **end**

13   $\mathcal{C}^{\text{surrogate}} \leftarrow \emptyset$;

14   **for** $k \leftarrow 1, ..., K$;          ◁ Find surrogate classes

15   **do**

16      $\mathcal{C}^{\text{surrogate}} \leftarrow \mathcal{C}^{\text{surrogate}} \bigcup \underset{t \in \{1,...,|\mathcal{C}^{\text{base}}|\}}{\text{Top-}R} d(\boldsymbol{c}_k, \boldsymbol{c}'_t)$

17   **end**

18   $\tilde{R} \leftarrow |\mathcal{C}^{\text{surrogate}}|$;

19   $\boldsymbol{d}' \leftarrow (d(\boldsymbol{z}, \boldsymbol{c}'_1), ..., d(\boldsymbol{z}, \boldsymbol{c}'_{\tilde{R}}))$ ;      ◁ Compute surrogate representation for query sample

20   **for** $k \leftarrow 1, ..., K$;          ◁ Compute surrogate representations for support samples

21   **do**

22      $\boldsymbol{d}'_k \leftarrow (d(\boldsymbol{c}_k, \boldsymbol{c}'_1), ..., d(\boldsymbol{c}_k, \boldsymbol{c}'_{\tilde{R}}))$;

23      $d''_k \leftarrow d(\boldsymbol{d}', \boldsymbol{d}'_k)$

24   **end**

25   $\tilde{d} \leftarrow \beta \frac{\boldsymbol{d}}{||\boldsymbol{d}||_1} + \gamma \frac{\boldsymbol{d}''}{||\boldsymbol{d}''||_1}$ ;      ◁ Compute final criterion

26   **return** $\tilde{d}$

---

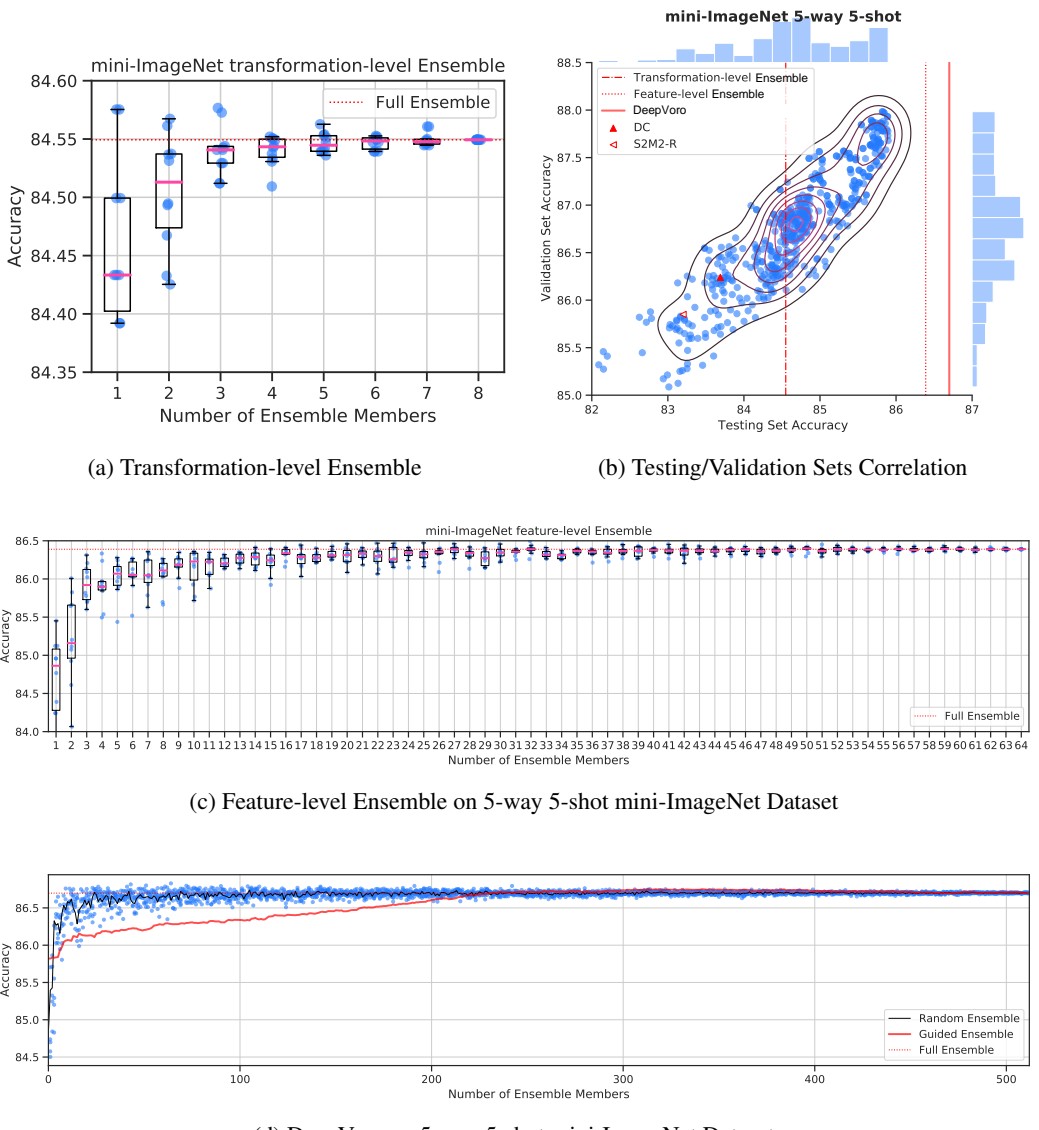

(a) Transformation-level Ensemble

(b) Testing/Validation Sets Correlation

(c) Feature-level Ensemble on 5-way 5-shot mini-ImageNet Dataset

(d) DeepVoro on 5-way 5-shot mini-ImageNet Dataset

Figure F.2: Three levels of ensemble and the correlation between testing and validation sets with different configurations in the configuration pool.

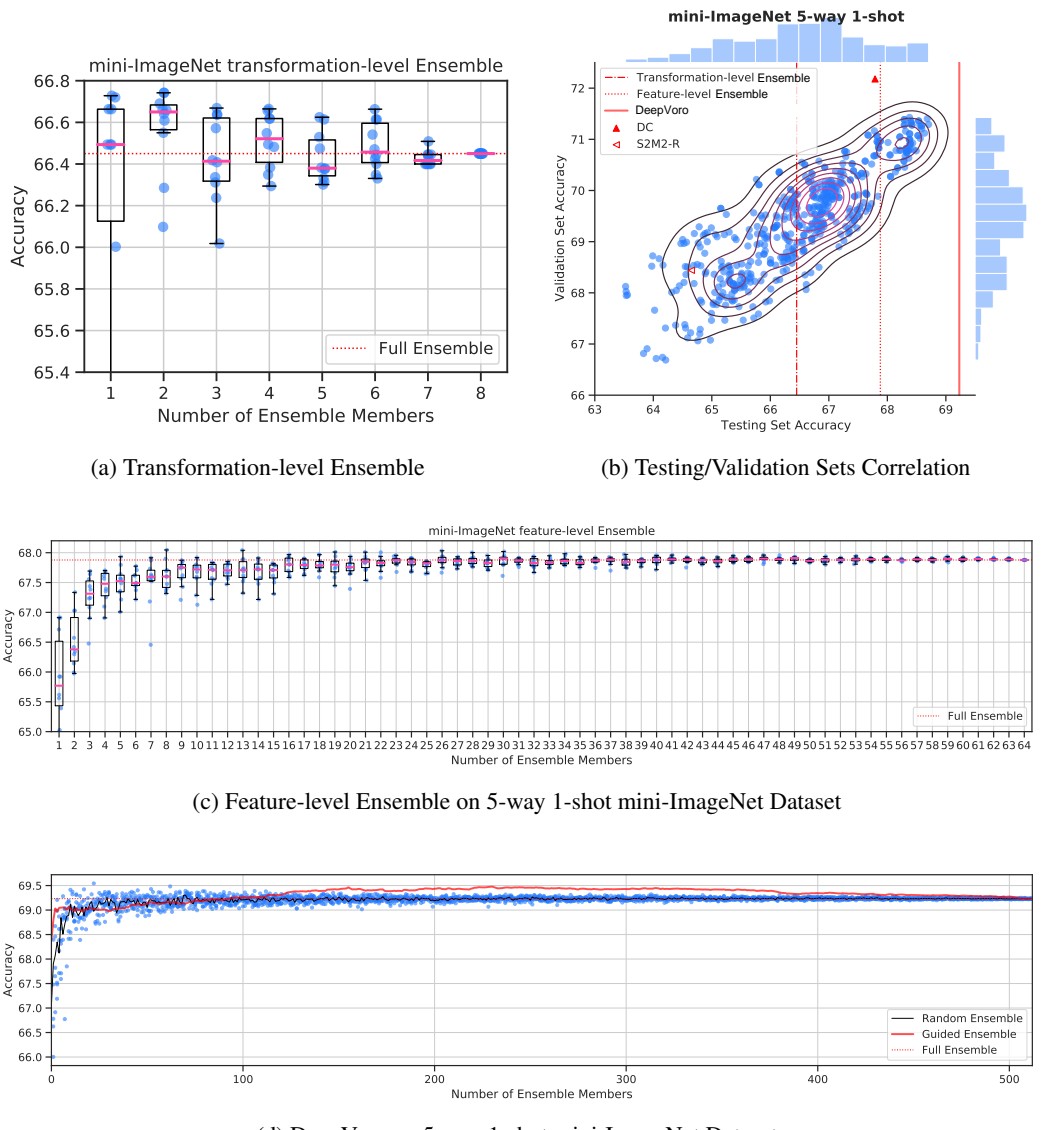

(a) Transformation-level Ensemble

(b) Testing/Validation Sets Correlation

(c) Feature-level Ensemble on 5-way 1-shot mini-ImageNet Dataset

(d) DeepVoro on 5-way 1-shot mini-ImageNet Dataset

Figure F.3: Three levels of ensemble and the correlation between testing and validation sets with different configurations in the configuration pool.

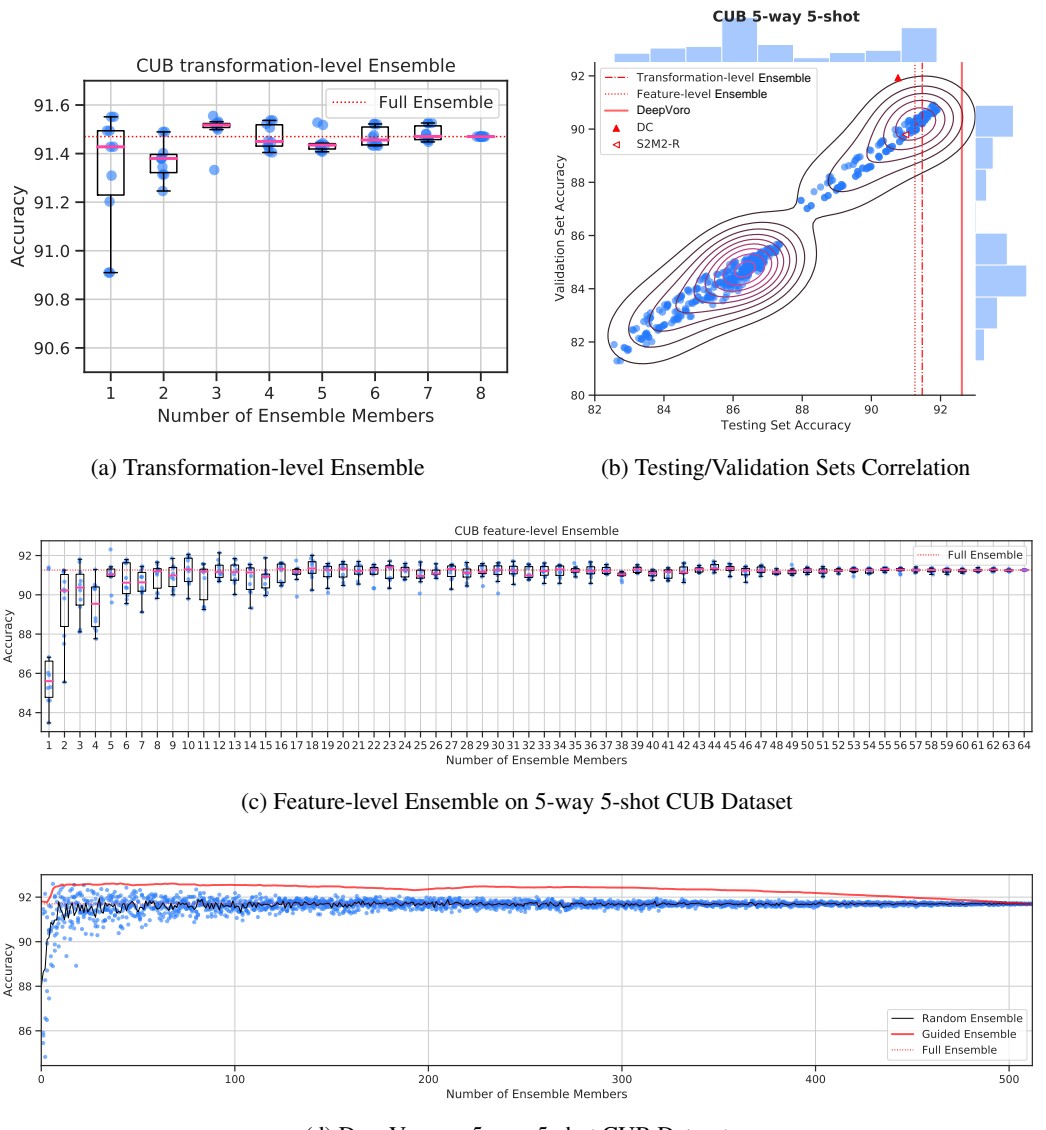

(a) Transformation-level Ensemble

(b) Testing/Validation Sets Correlation

(c) Feature-level Ensemble on 5-way 5-shot CUB Dataset

(d) DeepVoro on 5-way 5-shot CUB Dataset

Figure F.4: Three levels of ensemble and the correlation between testing and validation sets with different configurations in the configuration pool.

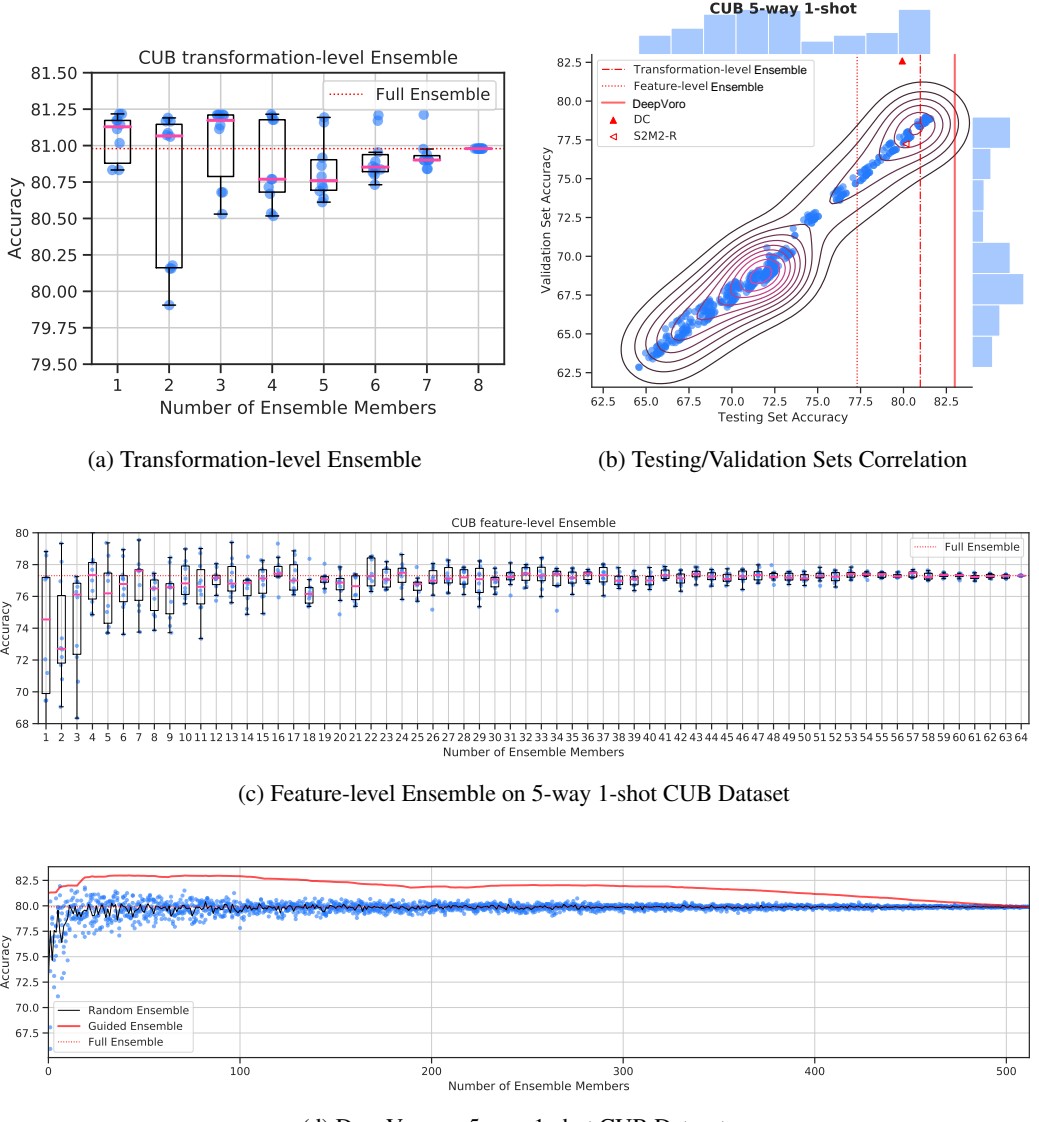

(a) Transformation-level Ensemble

(b) Testing/Validation Sets Correlation

(c) Feature-level Ensemble on 5-way 1-shot CUB Dataset

(d) DeepVoro on 5-way 1-shot CUB Dataset

Figure F.5: Three levels of ensemble and the correlation between testing and validation sets with different configurations in the configuration pool.

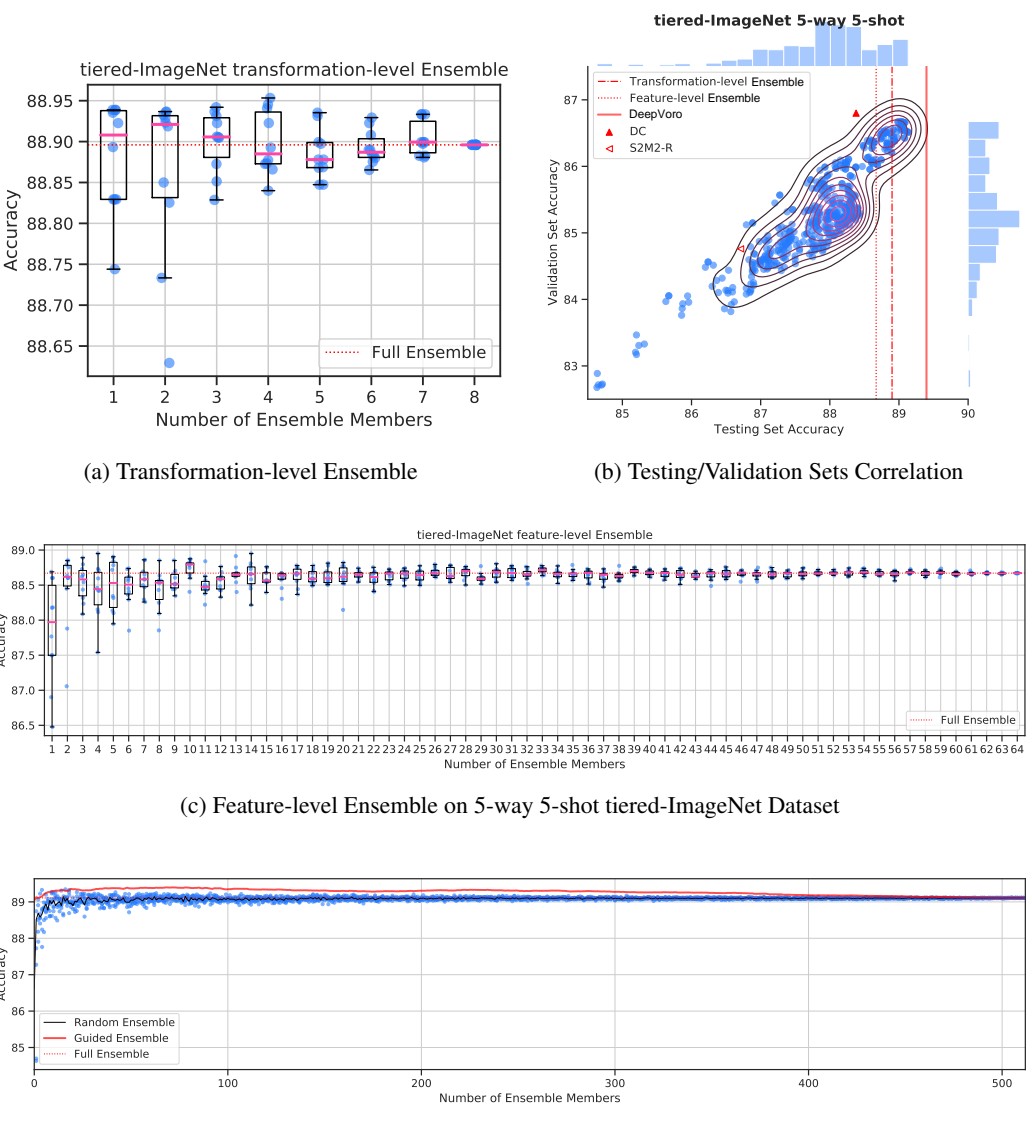

(a) Transformation-level Ensemble

(b) Testing/Validation Sets Correlation

(c) Feature-level Ensemble on 5-way 5-shot tiered-ImageNet Dataset

(d) DeepVoro on 5-way 5-shot tiered-ImageNet Dataset

Figure F.6: Three levels of ensemble and the correlation between testing and validation sets with different configurations in the configuration pool.

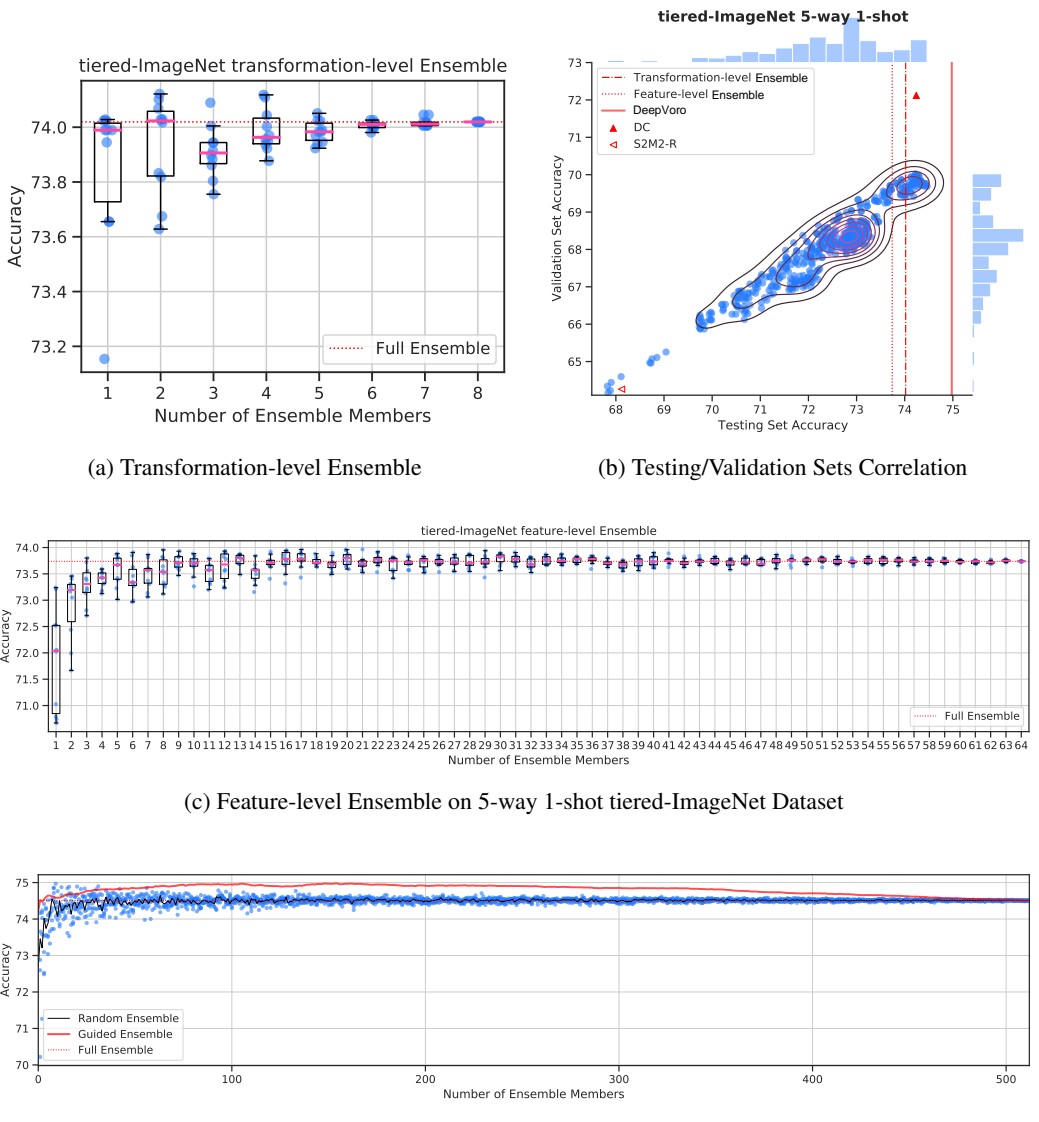

(a) Transformation-level Ensemble

(b) Testing/Validation Sets Correlation

(c) Feature-level Ensemble on 5-way 1-shot tiered-ImageNet Dataset

(d) DeepVoro on 5-way 1-shot tiered-ImageNet Dataset

Figure F.7: Three levels of ensemble and the correlation between testing and validation sets with different configurations in the configuration pool.

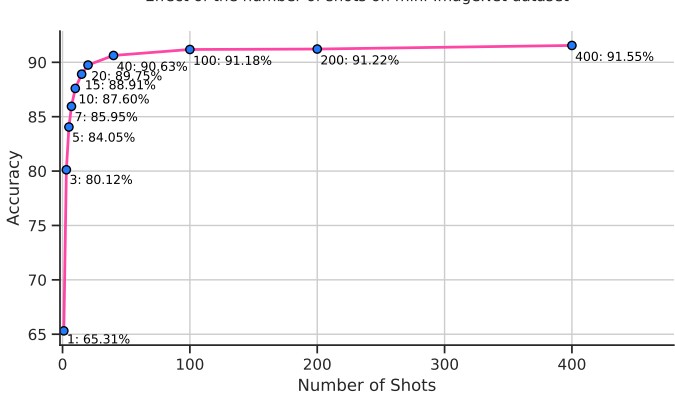

Figure G.8: The accuracy of VD with increasing number of shots on mini-ImageNet dataset.

## G  DEEPVORO++: FURTHER IMPROVEMENT OF FSL VIA SURROGATE REPRESENTATION

### G.1  EXPERIMENTAL SETUP AND IMPLEMENTATION DETAILS

In this section, we introduce another layer of heterogeneity, that is, geometry-level, that exists in our surrogate representation. In Definition 3.4, increasing $R$ will enlarge the degree of locality when searching for the top-$R$ surrogate classes. In equation (8), if we set $\gamma = 1$ then increasing $\beta$ will make the model rely more on the feature representation and less on the surrogate representation. In order to weigh up $R$ and $\beta$, we perform a grid search for different combinations of $R$ and $\beta$ on the validation set, as shown in Figure K.13, K.14, and K.15. For each $R$, we select the $\beta$ that gives rise to the best result on the validation set, and use this $(R, \beta)$ on the testing set, resulting in 10 such pairs in total. So there are 10 models in the geometry-level heterogeneity, standing for different degrees of locality. In conjunction with feature-level (64 kinds of augmentations) and transformation-level (here only the top-2 best transformations are used) heterogeneities, now there are 1280 different kinds of configurations in our configuration pool that will be used by the CCVD model. In conclusion, there are overall $512 + 1280 = 1792$ configurations for a few-shot episode. Generating $\sim 1800$ ensemble candidates is nearly intractable for parametric methods like logistic regression or cosine classifier, which may consume e.g. months for thousands of episodes. However, the VD model is nonparametric and highly efficient, making it empirically possible to collect all the combinations and integrate them all together via CCVD. The complete algorithm for the computation of surrogate representation is shown in Algorithm 2.

### G.2  RESULTS

The heatmaps for different $(R, \beta)$ pairs on testing/validation sets are shown in Figure K.13 for mini-ImageNet, in Figure K.14 for CUB, and in Figure K.15 for tiered-ImageNet, respectively. Basically, the testing and validation set follow the same pattern. When $R$ is small, i.e. only a small number of base classes are used for surrogate, then a higher weight should be placed on feature representation. With a fixed $\beta$, increasing $R$ beyond a certain threshold will potentially cause a drop in accuracy, probably because the meaningful similarities is likely to be overwhelmed by the signals from the large volume of irrelevant base classes.

Table G.5: Ablation study of DeepVoro++'s performance with different levels of ensemble. The number of ensemble members are given in parentheses.

| Methods | Feature-level | Transformation-level | Geometry-level | mini-ImageNet | CUB | tiered-ImageNet |
|---|---|---|---|---|---|---|
| No Ensemble | ✗ | ✗ | ✗ | $65.37 \pm 0.44$ | $78.57 \pm 0.44$ | $72.83 \pm 0.49$ |
| Vanilla Ensemble (20) | ✗ | ✔ | ✔ | $68.38 \pm 0.46$ | $80.70 \pm 0.45$ | $74.48 \pm 0.50$ |
| Vanilla Ensemble (64) | ✔ | ✗ | ✗ | $70.95 \pm 0.46$ | $81.04 \pm 0.44$ | $74.87 \pm 0.49$ |
| Vanilla Ensemble (1280) | ✔ | ✔ | ✔ | $71.24 \pm 0.46$ | $81.18 \pm 0.44$ | $74.75 \pm 0.49$ |
| Random Ensemble (1280) | ✔ | ✔ | ✔ | $\mathbf{71.34 \pm 0.46}$ | $81.98 \pm 0.43$ | $75.07 \pm 0.48$ |
| Guided Ensemble (1280) | ✔ | ✔ | ✔ | $71.30 \pm 0.46$ | $\mathbf{82.95 \pm 0.43}$ | $\mathbf{75.38 \pm 0.48}$ |

As shown in Table 3 and G.5, DeepVoro++ further improves upon DeepVoro for 5-way 1-shot FSL by $1.82\%$ and $0.4\%$ on mini-ImageNet and tiered-ImageNet, respectively, and is comparable with DeepVoro on CUB dataset ($82.95\%$ vs. $82.99\%$). Notably, on 5-shot FSL, DeepVoro++ usually causes a drop of accuracy from DeepVoro. To inspect the underlying reason for this behavior, we apply VD on 5-way $K$-shot FSL with $K$ increasing from 1 to 400 and report the average accuracy in Figure G.8. It can be observed that, in extreme low-shot learning, i.e. $K \in [1, 5]$, simply adding one shot makes more prominent contribution to the accuracy, suggesting that the centers obtained from 5-shot samples are much better that those from only 1 sample, so there is no necessity to resort to surrogate representation for multi-shot FSL and we only adopt DeepVoro for 5-shot episodes in the remaining part of this paper.

Ablation study of DeepVoro++ with different levels of ensemble is shown in Table G.5, Figure G.9, G.10, and G.11. All three layers of heterogeneities collaboratively contribute towards the final result. The fully-fledged DeepVoro++ establishes new state-of-the-art performance on all three datasets for 1-shot FSL.

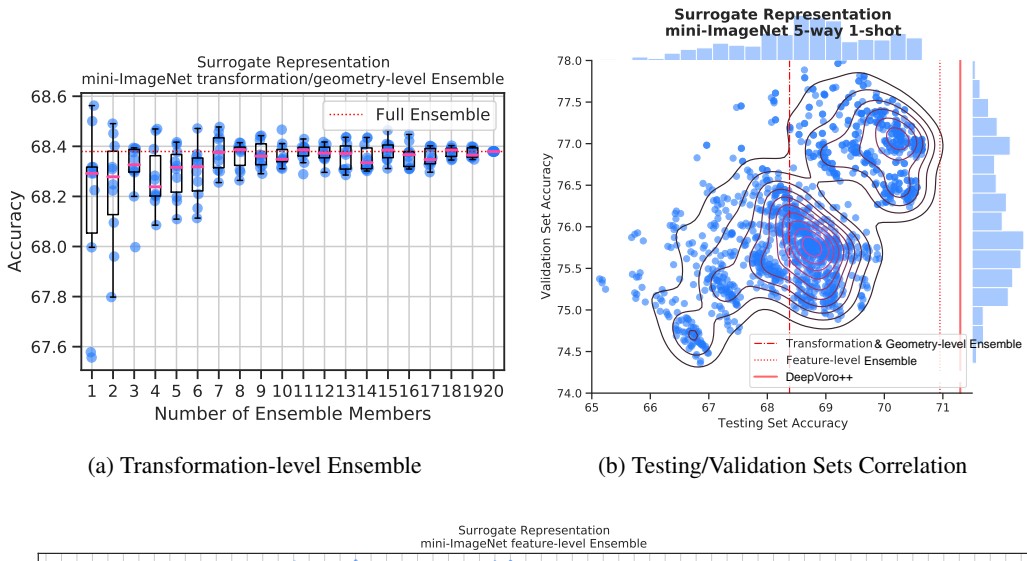

(a) Transformation-level Ensemble       (b) Testing/Validation Sets Correlation

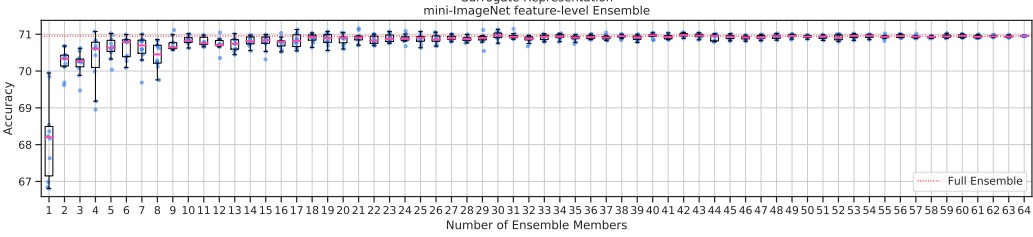

(c) Feature-level Ensemble on 5-way 1-shot mini-ImageNet Dataset

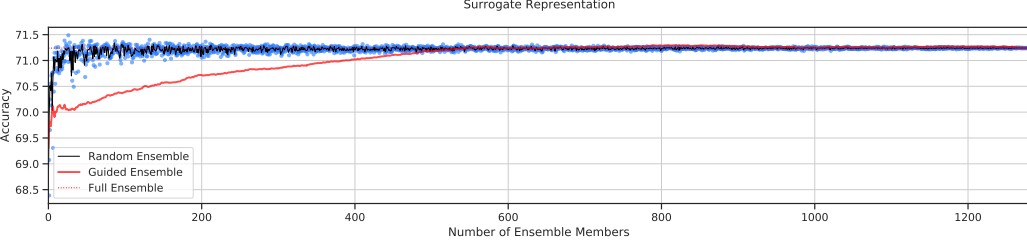

(d) DeepVoro++ on 5-way 1-shot mini-ImageNet Dataset

Figure G.9: Three levels of ensemble and the correlation between testing and validation sets with different configurations in the configuration pool.

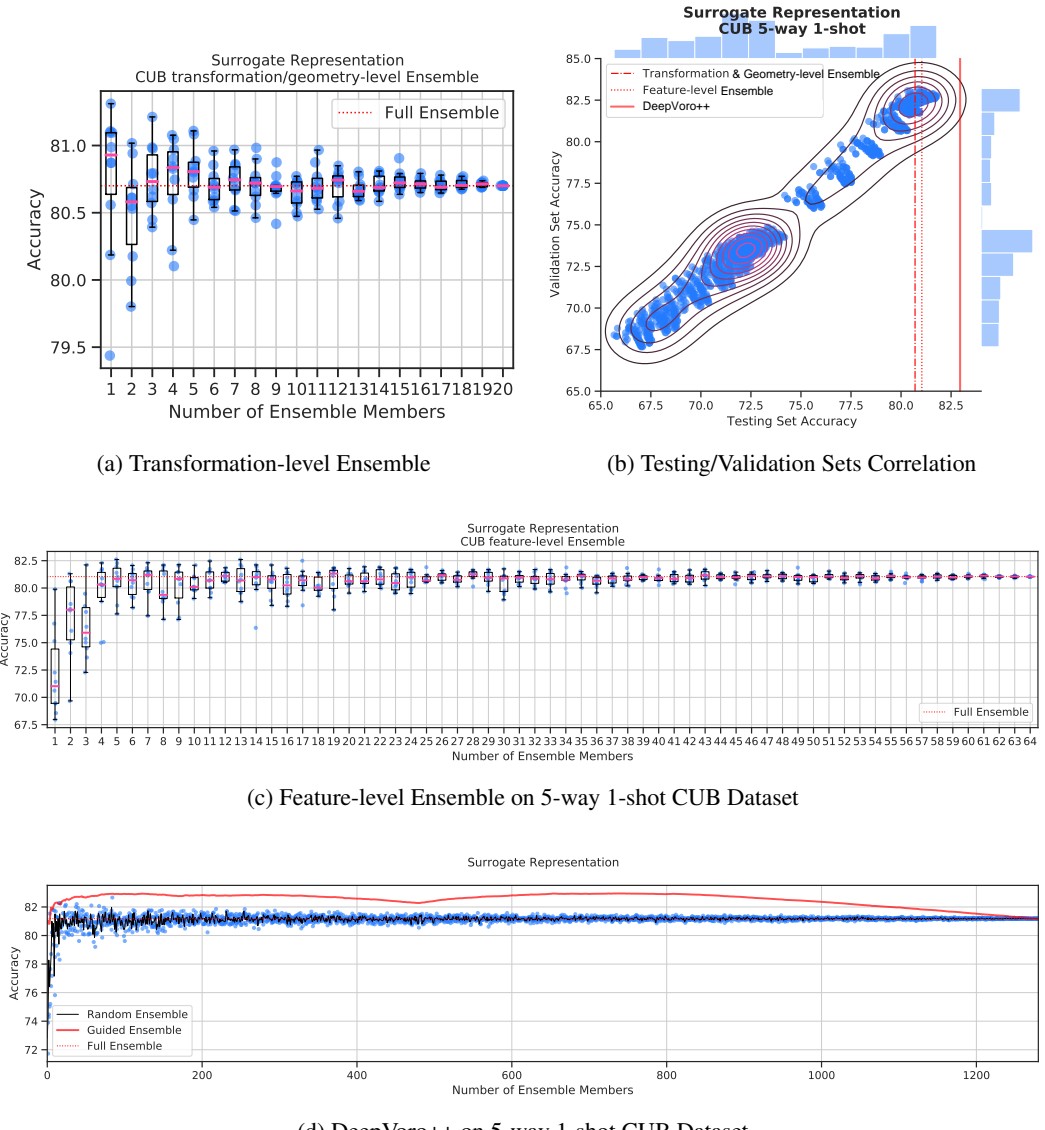

(a) Transformation-level Ensemble

(b) Testing/Validation Sets Correlation

(c) Feature-level Ensemble on 5-way 1-shot CUB Dataset

(d) DeepVoro++ on 5-way 1-shot CUB Dataset

Figure G.10: Three levels of ensemble and the correlation between testing and validation sets with different configurations in the configuration pool.

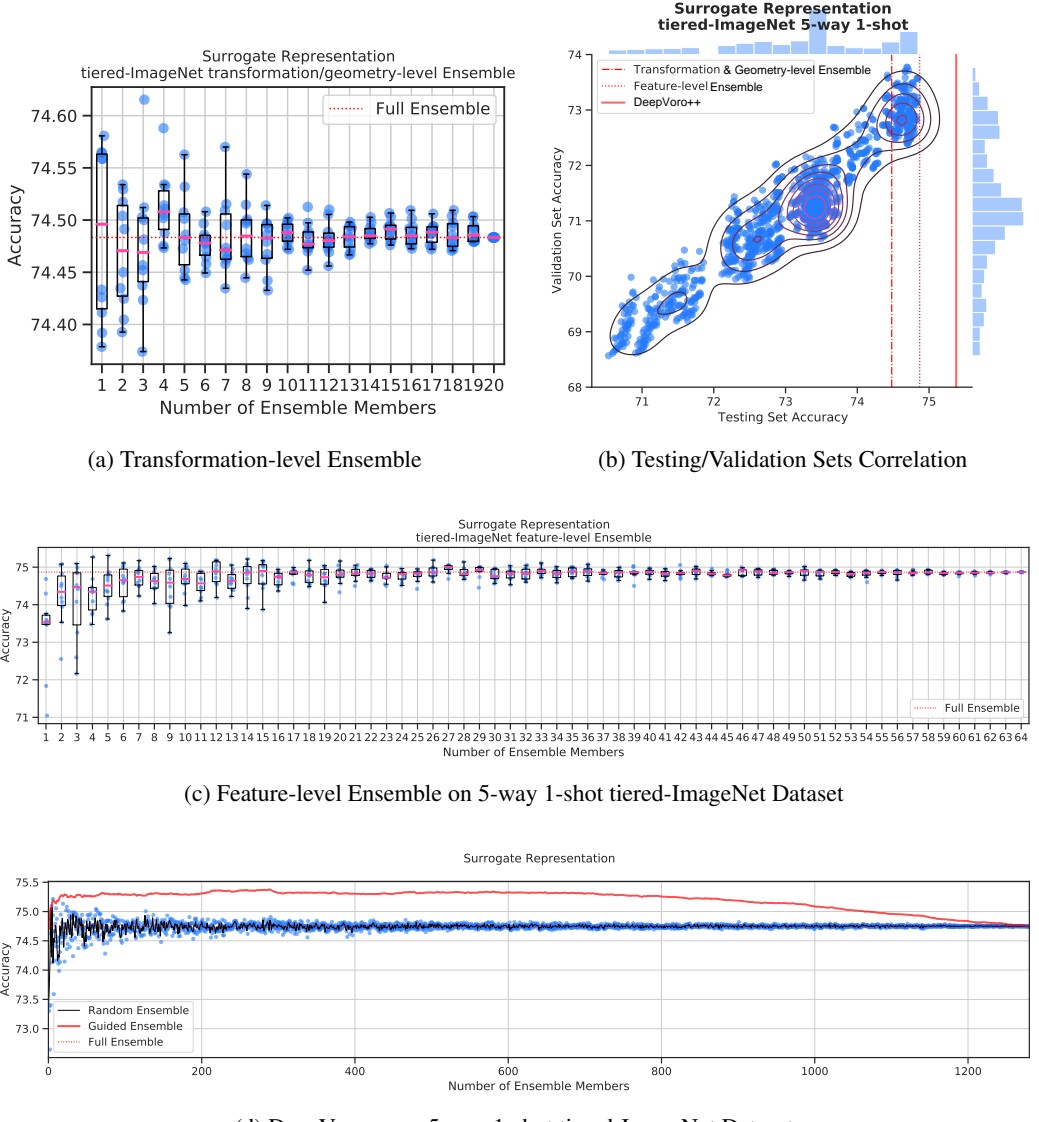

(a) Transformation-level Ensemble

(b) Testing/Validation Sets Correlation

(c) Feature-level Ensemble on 5-way 1-shot tiered-ImageNet Dataset

(d) DeepVoro++ on 5-way 1-shot tiered-ImageNet Dataset

Figure G.11: Three levels of ensemble and the correlation between testing and validation sets with different configurations in the configuration pool.

## H    EXPERIMENTS WITH DIFFERENT BACKBONES

### H.1    IMPLEMENTATION DETAILS

In order to test the robustness of DeepVoro/DeeoVoro++ with various deep learning architectures, we downloaded the trained models[2] used by Wang et al. (2019). We evaluated DC, S2M2_R, DeepVoro, and DeepVoro++ using the same random seed. The results are obtained by running 500 episodes and the average accuracy as well as 95% confidence intervals are reported.

### H.2    EXPERIMENTAL RESULTS

Table H.6: Comparison of FSL algorithms with different network architectures. WRN-28-10§ was trained with rotation loss and MixUp loss (Mangla et al., 2020) instead of using ordinary softmax loss (WRN-28-10†).

| Methods | WRN-28-10§ | | WRN-28-10† | |
|---|---|---|---|---|
| | 1-shot | 5-shot | 1-shot | 5-shot |
| DC | $67.79 \pm 0.45$ | $83.69 \pm 0.31$ | $62.09 \pm 0.95$ | $78.47 \pm 0.67$ |
| S2M2_R | $64.65 \pm 0.45$ | $83.20 \pm 0.30$ | $61.11 \pm 0.92$ | $79.83 \pm 0.64$ |
| **DeepVoro** | $69.48 \pm 0.45$ | $\mathbf{86.75 \pm 0.28}$ | $62.26 \pm 0.94$ | $\mathbf{82.02 \pm 0.63}$ |
| **DeepVoro++** | $\mathbf{71.30 \pm 0.46}$ | – | $\mathbf{65.01 \pm 0.98}$ | – |
| | **DenseNet-121** | | **ResNet-34** | |
| | 1-shot | 5-shot | 1-shot | 5-shot |
| DC | $62.68 \pm 0.96$ | $79.96 \pm 0.60$ | $59.10 \pm 0.90$ | $74.95 \pm 0.67$ |
| S2M2_R | $60.33 \pm 0.92$ | $80.33 \pm 0.62$ | $58.92 \pm 0.92$ | $77.99 \pm 0.64$ |
| **DeepVoro** | $60.66 \pm 0.91$ | $\mathbf{82.25 \pm 0.59}$ | $61.61 \pm 0.92$ | $\mathbf{81.81 \pm 0.60}$ |
| **DeepVoro++** | $\mathbf{65.18 \pm 0.95}$ | – | $\mathbf{64.65 \pm 0.96}$ | – |
| | **ResNet-18** | | **ResNet-10** | |
| | 1-shot | 5-shot | 1-shot | 5-shot |
| DC | $60.20 \pm 0.96$ | $75.59 \pm 0.69$ | $59.01 \pm 0.92$ | $74.27 \pm 0.69$ |
| S2M2_R | $59.57 \pm 0.93$ | $78.69 \pm 0.69$ | $57.59 \pm 0.92$ | $77.10 \pm 0.67$ |
| **DeepVoro** | $61.50 \pm 0.93$ | $\mathbf{81.58 \pm 0.64}$ | $58.34 \pm 0.93$ | $\mathbf{79.05 \pm 0.63}$ |
| **DeepVoro++** | $\mathbf{64.79 \pm 0.97}$ | – | $\mathbf{61.75 \pm 0.95}$ | – |
| | **MobileNet** | | **Conv-4** | |
| | 1-shot | 5-shot | 1-shot | 5-shot |
| DC | $59.41 \pm 0.91$ | $76.07 \pm 0.66$ | $49.32 \pm 0.87$ | $62.89 \pm 0.71$ |
| S2M2_R | $58.36 \pm 0.93$ | $76.75 \pm 0.68$ | $45.19 \pm 0.87$ | $64.56 \pm 0.74$ |
| **DeepVoro** | $60.91 \pm 0.93$ | $\mathbf{80.14 \pm 0.65}$ | $48.47 \pm 0.86$ | $\mathbf{65.86 \pm 0.73}$ |
| **DeepVoro++** | $\mathbf{63.37 \pm 0.95}$ | – | $\mathbf{52.15 \pm 0.98}$ | – |

On Wide residual networks (WRN-28-10) (Zagoruyko & Komodakis, 2016), Residual networks (ResNet-10/18/34) (He et al., 2016), Dense convolutional networks (DenseNet-121) (Huang et al., 2017), and MobileNet (Howard et al., 2017), DeepVoro/DeepVoro++ shows a consistent improvement upon DC and S2M2_R. Excluding DeepVoro/DeepVoro++, there is no such a method that is always better for both 5-shot and 1-shot FSL. Generally, DC is expert in 1-shot while S2M2_2 favors 5-shot. According to Table 3, we do not apply DeepVoro++ on 5-shot FSL since DeepVoro usually outperforms DeepVoro++ with more shots available.

## I    EXPERIMENTS WITH DIFFERENT TRAINING PROCEDURES

### I.1    EXPERIMENTAL SETUP AND IMPLEMENTATION DETAILS

Our geometric space partition model is built on top of a pretrained feature extractor, and the quality of the feature extractor will significantly affect the downstream FSL (Mangla et al., 2020). Here we used another two feature extractors trained with different schemes. (1) Manifold Mixup training employs an additional Mixup loss that interpolates the data and the label simultaneously and can

---

[2]downloaded from `https://github.com/mileyan/simple_shot`

help deep neural network generalize better. (2) Rotation loss is widely used especially in self-supervised learning in which the network learns to predict the degree by which an image is rotated. We downloaded the corresponding pretrained models used by Mangla et al. (2020) and Yang et al. (2021) and evaluate the four methods by 500 episodes.

## I.2 RESULTS

Table I.7: Comparison of performance with different meta-training procedures.

| Methods | Self-supervision w/ Rotation Loss | | Manifold Mixup | |
|---------|-----------------|-----------------|-----------------|-----------------|
| | 1-shot | 5-shot | 1-shot | 5-shot |
| DC | $66.43 \pm 0.86$ | $82.61 \pm 0.62$ | $62.61 \pm 0.90$ | $78.62 \pm 0.68$ |
| S2M2_R | $58.33 \pm 0.96$ | $79.26 \pm 0.66$ | $48.11 \pm 0.96$ | $72.74 \pm 0.74$ |
| **DeepVoro** | $68.80 \pm 0.86$ | $\mathbf{85.70 \pm 0.58}$ | $65.00 \pm 0.93$ | $\mathbf{83.19 \pm 0.65}$ |
| **DeepVoro++** | $\mathbf{69.23 \pm 0.89}$ | – | $\mathbf{65.25 \pm 0.93}$ | – |

As shown in Table I.7, DeepVoro/DeepVoro++ achieves best results for both rotation loss and Mixup loss. Interestingly, there is a substantial gap between the two training schemes when they are used out-of-the-box for downstream FSL ($\triangle$accuracy $= 10.22\%$ for 1-shot and $\triangle$accuracy $= 6.52\%$ for 5-shot), but after DeepVoro/DeepVoro++, this gap becomes narrowed ($\triangle$accuracy $= 3.98\%$ for 1-shot and $\triangle$accuracy $= 2.51\%$ for 5-shot), suggesting the strength of DeepVoro/DeepVoro++ to make the most of the pretrained models

## J CROSS DOMAIN FEW-SHOT LEARNING

### J.1 EXPERIMENTAL SETUP AND IMPLEMENTATION DETAILS

Cross-domain FSL is more challenging than FSL in which base classes and novel classes come from essentially distinct domains. To examine the ability of our method for cross-domain FSL, we apply the feature extractor trained on mini-ImageNet (CUB) on the few-shot data in CUB (mini-ImageNet) for coarse-to-fine (fine-to-coarse) domain shifting.

### J.2 RESULTS

Table J.8: Comparison of performance on cross-domain FSL.

| Methods | CUB $\Rightarrow$ mini-ImageNet | | mini-ImageNet $\Rightarrow$ CUB | |
|---------|-----------------|-----------------|-----------------|-----------------|
| | 1-shot | 5-shot | 1-shot | 5-shot |
| DC | $46.25 \pm 0.93$ | $62.99 \pm 0.81$ | $54.64 \pm 0.87$ | $\mathbf{72.83 \pm 0.71}$ |
| S2M2_R | $41.15 \pm 0.84$ | $58.09 \pm 0.79$ | $49.01 \pm 0.88$ | $69.99 \pm 0.71$ |
| **DeepVoro** | $46.15 \pm 0.90$ | $\mathbf{64.60 \pm 0.80}$ | $49.03 \pm 0.87$ | $72.30 \pm 0.74$ |
| **DeepVoro++** | $\mathbf{47.83 \pm 0.97}$ | – | $\mathbf{54.88 \pm 0.92}$ | – |

Basically, DeepVoro/DeepVoro++ is more stable than the other two method with a shifting domain, especially on fine-to-coarse FSL (CUB to mini-ImageNet), with an improvement of $6.68\%$ for 1-shot and $6.51\%$ for 5-shot than S2M2_R, and is comparable with DC on coarse-to-fine FSL (mini-ImageNet to CUB).

## K ADDITIONAL FIGURES

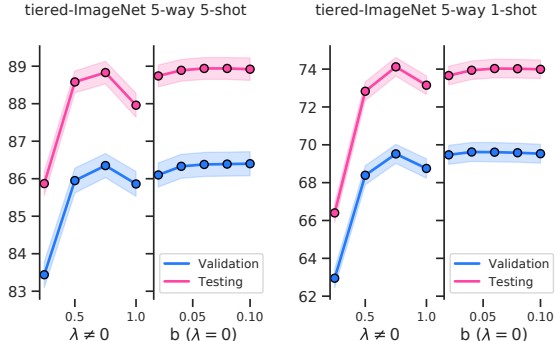

Figure K.12: The 5-way few-shot accuracy of VD with different $\lambda$ and $b$ values on tiered-ImageNet datasets.

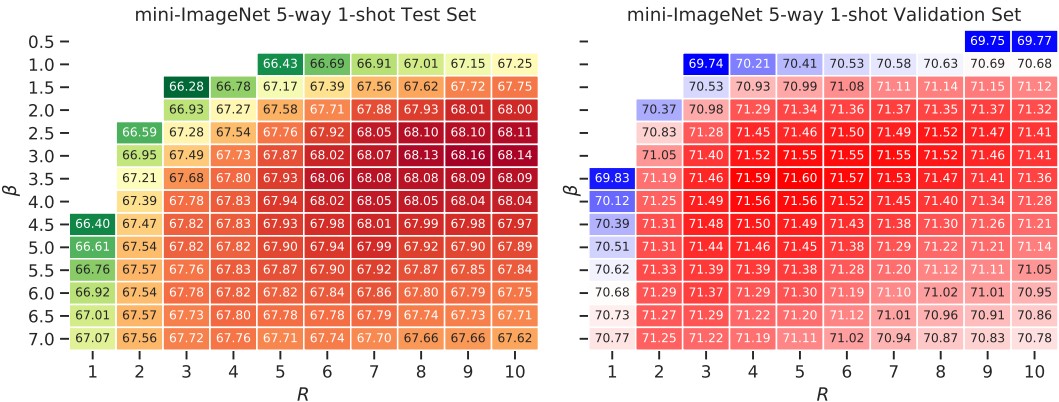

Figure K.13: The 5-way 1-shot accuracy with different $\beta$ and $R$ values on mini-ImageNet testing (left) and validation (right) datasets.

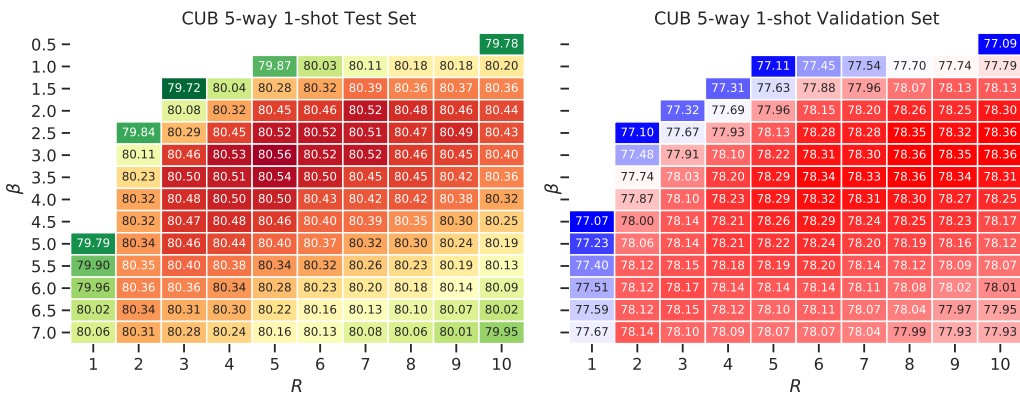

Figure K.14: The 5-way 1-shot accuracy with different $\beta$ and $R$ values on CUB testing (left) and validation (right) datasets.

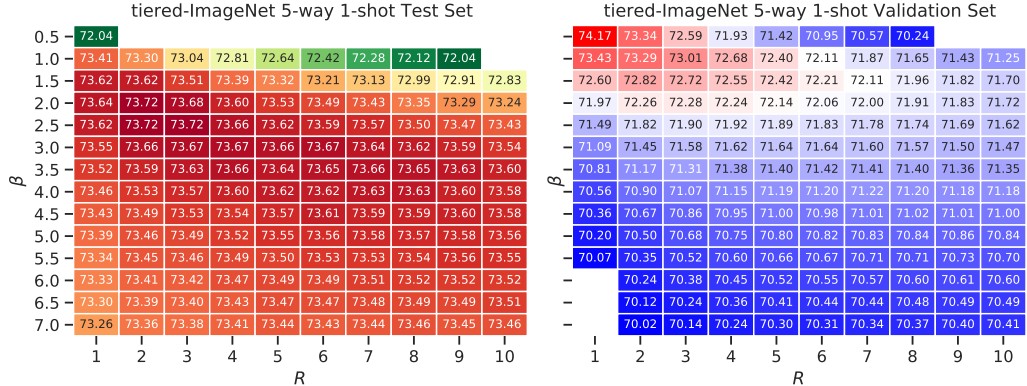

Figure K.15: The 5-way 1-shot accuracy with different $\beta$ and $R$ values on tiered-ImageNet testing (left) and validation (right) datasets.

Episode Accuracy as a Function of Geometric Variance

Figure K.16: Outlier Analysis. In order to investigate the resistance to outlier for various methods, we here define **Geometric Variance** ($GV$) as a reflection of the possibility that a support set contains an outlier, due to the difficulty of inferring out-of-distribution sample from merely 1 or 5 samples. Formally, for a support set $\mathcal{S} = \{(\boldsymbol{z}_i, y_i)\}_{i=1}^{K \times N}$, its Geometric Variance is defined as $GV(\mathcal{S}) = \frac{1}{K}\sum_{k=1}^{K} \frac{1}{\binom{N}{2}} \sum_{i \in \{1,\dots,N\}, j \in \{1,\dots,N\}} \|\boldsymbol{z}_i - \boldsymbol{z}_j\|_2$, measuring the average point-to-point distance in this support set. The larger $GV$ is, with higher probability $\mathcal{S}$ contains an outlier. For every episode in 2000 episodes from 5-way 5-shot mini-ImageNet data, $GV$ is computed as well as the episode accuracy. As shown in Figure K.16, very high $GV$ causes a significant decrease of episode accuracy, but our method DeepVoro is more resistant to the presence of outliers.