# OpenReview forum: "Few-shot Learning via Dirichlet Tessellation Ensemble"
_ICLR.cc/2022/Conference — ICLR 2022 Poster_

### Official Review · Reviewer_uC53 · 2021-11-02

**Correctness:** 3
**Technical Novelty And Significance:** 2
**Empirical Novelty And Significance:** 3
**Recommendation:** 6
**Confidence:** 3

**Main Review:**

The paper is written in a way that makes it extremely hard to read. The text is not sufficiently self-contained to follow the line of arguments, terms and methods are not always well-defined, and many side-tracks with questionable relevance are considered. Most importantly, the evaluated methods DeepVoro(++) are never defined in the method part.

Terminology is sometimes used in unusual ways, e.g. I would identify data augmentation rather with what is said under point 3 in section 1 than GANs and VAEs in the top of the page.

Figure 1 is confusing. Usually, when adding new classes, old ones are not supposed to be forgotten (cmp. left and center panel).

Some equations contain trivial errors, e.g. the optimization below (1) should be a minimization of distance.

Some presentations confuse unnecessarily, e.g. (3) where an offset/bias for the distance is introduced, but it is called weight. The text below uses Dirichlet tessellation as a special case of PD, whereas before (3) they are identified.

The datasets used for the comparisons should be described in the main paper, not the supplementary material.

The overall results look impressive, but the paper needs a restructuring such that the following obvious questions are answered: How exactly are the methods DeepVoro(++) defined, what is their baseline and what modifications have been made? How are these modifications motivated from theory or experiments? Which of these modifications or theoretical insights are the main contributions of the paper?

In order to address these questions, include the dataset descriptions and the most relevant ablative results, section 3 needs to be streamlined and less relevant theoretical considerations need to be removed or moved to the supplementary material.

**Summary Of The Paper:**

The paper proposes a CIVD-based approach to few-shot learning. CIVD, cluster-induced Voronoi diagrams, are a known technique that is used to categorize / describe different types of few-shot classifiers. In the experiment section DeepVoro(++) is shown to perform superior to other methods on three datasets.

**Summary Of The Review:**

The manuscript is technically overloaded but not self-contained. Important descriptions, in particular of the proposed methods, are missing. The experimental results look impressive, but it remains unclear what exactly has been used in the DeepVoro(++) methods.

Post-rebuttal:
most concerns have been addressed and the correctness has improved. The assessment has been raised accordingly.

---

> ### Author Response · Authors · 2021-11-19
> **Response to Reviewer uC53 (Q1-Q6)**
>
> Thanks for your time and efforts in reviewing this paper! We are sorry about the unnecessary confusion and misunderstanding. Here we would like to make some clarifications regarding your concerns.
>
> **Q1:** The text is not sufficiently self-contained and the evaluated methods DeepVoro(++) are never defined in the method part. \
> **A1:** In the *submitted version*, the terms DeepVoro(++) are defined in the Experiments section (not in the Methodology section). Table 1, in which DeepVoro appears, is used to summarize the experiments. We are sorry if this causes confusion. Basically, DeepVoro is our method that uses Cluster-to-cluster Voronoi Diagram (CCVD) as the underlying geometric structure, which is promoted to DeepVoro++ when surrogate representation is also incorporated. In *revised version*, we have highlighted the definitions of DeepVoro--/DeepVoro/DeepVoro++ in Introduction, Methodology, and Experiments (see red text). Furthermore, we have added Table 1 and Table A.2 that give a clear comparison of different methods along with their geometric structures. To improve the self-containment of the paper, we have added Table A.1 summarizing and comparing VD, PD, CIVD, and CCVD, four geometric structures discussed in the paper.
>
> **Q2:** I would identify data augmentation rather than GANs and VAEs. \
> **A2:** In FSL, GANs and VAEs can be used as a way of data augmentation. You may find a similar description in [1]: "Most methods use the idea of GANs and VAEs to generate samples or features to *augment* the training set".
>
> **Q3:** In Figure 1, when adding new classes, old ones are not supposed to be forgotten. \
> **A3:** Yes, you are right that base classes are not forgotten when classifying novel classes. However, the base classes will not be involved when making predictions for novel classes. Taking cross-domain FSL as a good example, where the base classes are from mini-Imagenet and novel classes are from CUB. When making predictions we only discriminate between different species of *birds* without considering *other classes* (e.g. dogs) from mini-Imagenet, and this does not mean that the latter are forgotten.
>
> **Q4:** The optimization below (1) should be a minimization of distance. \
> **A4:** Thanks for this correction. We have fixed this typo (see green text).
>
> **Q5:** In equation (3), an offset/bias for the distance is introduced, but it is called weight. The text below uses Dirichlet tessellation as a special case of PD, whereas before (3) they are identified. \
> **A5:** Thanks for raising concerns about the terminology. However, the terms here are used in a way that is consistent with the computational geometry literature. The "offset/bias" is called "weight" because this is how it's named when Power Diagram was first proposed in 1987, see page 78 in [2]: "Each $p \in M$ has assigned an individual real number $w(p)$, the *weight* of $p$." In Definition 3.1, *sectional* Dirichlet Tessellation (not Dirichlet Tessellation) is first defined, which is a general form of Dirichlet Tessellation, then it collapses to Dirichlet Tessellation. In the *revised version*, to avoid possible misunderstanding, we discard the use of the term "Dirichlet Tessellation" throughout the paper, and refer to it all as Voronoi Diagram. The title is also modified to be "**Few-shot Learning as Cluster-induced Voronoi Diagrams: A Geometric Approach**".
>
> **Q6:** The datasets used for the comparisons should be described in the main paper, not in the supplementary material. \
> **A6:** Thanks for your suggestion. We described the datasets in the Appendix because two ICLR'21 papers [3] and [4] did so. Following your suggestion, in the *revised version*, we have added Table 2 to the main text summarizing the datasets used in the paper.

---

> > ### Author Response · Authors · 2021-11-19
> > **Response to Reviewer uC53 (Q7-Q12)**
> >
> > **Q7:** How exactly are the methods DeepVoro(++) defined? \
> > **A7:** In the submitted version, we didn't mention the description of DeeoVoro/DeepVoro++ until the experiment section, sorry if this causes confusion. Basically, DeepVoro is our method that uses Cluster-to-cluster Voronoi Diagram (CCVD) as the underlying geometric structure, which is then promoted to DeepVoro++ when surrogate representation is also incorporated. In the revised version, we have highlighted the definitions of DeepVoro--/DeepVoro/DeepVoro++ in Introduction, Methodology, and Experiments (see red text). Furthermore, we have added Table 1 and Table A.2 that give a clear comparison of different methods along with their geometric structures.
> >
> > **Q8:** What is their baseline and what modifications have been made? \
> > **A8:** As discussed in the Introduction, the ProtoNet model is geometrically the simplest Voronoi Diagram (VD) structure. Next, we improve the basic VD by two advanced geometric structures CIVD (Sec. 3.2) and CCVD (Sec. 3.4). Based on CIVD we propose DeepVoro-- (Sec. 3.2); based on CCVD we propose DeepVoro (Sec. 3.4). When surrogate representation (Sec. 3.3) is further incorporated, DeepVoro is then promoted to DeepVoro++.
> >
> > **Q9:** How are these modifications motivated by theory or experiments? \
> > **A9:** These modifications are mainly motivated by theory from computational geometry. As discussed in the Introduction, VD (the underlying geometric structure of ProtoNet) is the simplest but not necessarily the optimal geometric structure for FSL. This motivates us to explore more advanced forms of VD, i.e. CIVD/CCVD.
> >
> > **Q10:** Which of these modifications or theoretical insights are the main contributions of the paper? \
> > **A10:** Please let us summarize the main contributions of the paper: \
> > <1> The first work that reveals the underlying geometric structures for a number of prevailing FSL methods. \
> > <2> Proposes to use more advanced geometric structures (CIVD and CCVD) for better FSL (DeepVoro--/DeepVoro). \
> > <3> Proposes a novel surrogate representation that significantly improves 1-shot FSL (DeepVoro++). \
> > <4> Achieved state-of-the-art results on all three benchmark datasets.
> >
> > **Q11:** Ablative results. \
> > **A11:** In the *submitted version*, the full ablation studies are shown in Appendix Table D.1, E.2, and F.3. Sorry for not making them easier to be found. In the *revised version*, we have added Table 4 showing DeepVoro--/DeepVoro/DeepVoro++'s performance with different levels of ensemble.
> >
> > **Q12:** Less relevant theoretical considerations need to be removed or moved to the supplementary material. \
> > **A12:** Thanks for your suggestion. We have removed the "High Dimension, Low Sample Size Data" section in Related Works, and moved Lemma 3.1 to the Appendix (to be Lemma B.1).
> >
> > Hope these would address your concerns, and please let us know if you have any further questions. Thanks!
> >
> > **References** \
> > [1] Free Lunch for Few-shot Learning: Distribution Calibration, ICLR 2021. \
> > [2] Power Diagrams: Properties, Algorithms and Applications, SIAM Journal on Computing 1987. \
> > [3] Attentional Constellation Nets for Few-Shot Learning, ICLR 2021. \
> > [4] Bayesian Few-Shot Classification, ICLR 2021.

---

> > > ### Comment · Reviewer_uC53 · 2021-11-25
> > > **Concerns addressed**
> > >
> > > Most of the listed concerns are addressed appropriately, thanks a lot for the clarifications.
> > >
> > > The clarity of presentation has improved a lot, but certain central aspects might be worth being repeated or summarized. The authors might want to consider including a summary similar to their A8 at a suitable location in the manuscript.
> > >
> > > Regarding moving from "tesselation" to Voronoi diagrams, this means a substantial change of focus / approach on a conceptual level. It is a bit difficult to determine its impact on the aspect of novelty, but it definitely improves readability. Still, it had been nice to see a discussion of the possibly reduced novelty in the rebuttal response.

---

> > > > ### Author Response · Authors · 2021-11-29
> > > > **Thanks for your feedback**
> > > >
> > > > Thanks for your suggestion! We will surely include this summary to the main text in the next version.
> > > >
> > > > We didn't mean to conceptually alter the approach, but wanted to keep the terms concise. In space subdivision, "tessellation" is the process, and "diagram" is the result. Although both Dirichlet Tessellation and Voronoi Diagram often mean the same geometric structure, Voronoi Diagram is probably a little more popular and easy to understand. So the main novelty remains the same: to interpret and then improve few-shot learning from a computational geometry point of view. We will add more details about the basics of computational geometry to make it more easily understood by the community. \
> > > > Thanks again for your feedback!

---

### Official Review · Reviewer_Qc9m · 2021-11-02

**Correctness:** 3
**Technical Novelty And Significance:** 3
**Empirical Novelty And Significance:** 3
**Recommendation:** 6
**Confidence:** 5

**Main Review:**

Strengths
1.	The new geometric point of view of FSL is novel and interesting. This paper makes a bridge between computational geometry and FSL.
2.	This approach achieves the new state-of-the-art performance over three standard datasets, including mini-ImageNet, CUB, and tiered-ImagenNet. This demonstrates its effectiveness.

Weaknesses

1.	The ablation study seems to be not enough. What is the performance of the proposed method if the ensemble strategy is removed? The reviewer did not find the corresponding experimental results.
2.	Usually, the ensemble strategy is time-cost. What is the computation time cost of the proposed methods, especially for Deep Voro++. This method with 1280 configurations seems to require a large amount of computation time.


**Summary Of The Paper:**

This paper provides a new geometric point of view for few-shot learning (FSL). In this view, the widely used ProtoNet can be regarded as a Dirichlet Tessellation (Voronoi Diagram)
in the feature space. Furthermore, the authors use the recent Cluster-induced Voronoi Diagram (CIVD) for FSL and propose an ensemble approach to achieve a stronger FSL model. Extensive experimental results on three standard benchmarks demonstrate the effectiveness of the proposed method.

**Summary Of The Review:**

The main idea is novel and interesting, and the experimental results demonstrate the superior performance of the proposed method. However, this paper lacks some ablation studies and time comparisons. In summary, the reviewer thinks this paper is marginally above the acceptance threshold.

---

> ### Author Response · Authors · 2021-11-19
> **Response to Reviewer Qc9m**
>
> Thanks to Reviewer Qc9m for your valuable comments! We address the concerns as follows.
>
> **Q1:** Ablation study -- what is the performance of the proposed method if the ensemble strategy is removed? \
> **A1:** For both DeepVoro/DeepVoro++, we remove the proposed CCVD integration, and show the average accuracy of 512/1280 individual VDs as follows:
>
> |  Methods   | mini-ImageNet |            |    CUB     |            | tiered-ImageNet |            |
> | ---------- | ------------  | ---------- | ---------- | ---------- | --------------  | ---------- |
> |            | 1-shot        | 5-shot     | 1-shot     | 5-shot     | 1-shot          | 5-shot     |
> | VDs        | 66.92±0.45    | 84.64±0.30 | 73.19±0.47 | 87.52±0.27 | 72.42±0.49      | 87.89±0.31 |
> | DeepVoro   | 69.48±0.45    | 86.75±0.28 | 82.99±0.43 | 92.62±0.22 | 74.98±0.48      | 89.40±0.29 |
> |            |               |            |            |            |                 |            |
> | VDs        | 68.68±0.46    | 84.28±0.31 | 73.91±0.49 | 89.77±0.25 | 73.17±0.50      | 86.37±0.34 |
> | DeepVoro++ | 71.30±0.46    | 85.40±0.30 | 82.95±0.43 | 91.21±0.23 | 75.38±0.48      | 87.25±0.33 |
>
> Clearly, when CCVD is removed, the average accuracy of individual VDs drops significantly, suggesting that, although the ensemble pool may contain some less accurate VDs, when integrated via CCVD, the overall accuracy surpasses every single VD. In the *submitted version*, the full ablation studies are shown in Appendix Table D.1, E.2, and F.3. In the *revised version*, we have added Table 4 showing DeepVoro--/DeepVoro/DeepVoro++'s performance with different levels of ensemble.
>
> **Q2:** What is the computation time of the proposed methods, especially for DeepVoro++? \
> **A2:** Since an individual VD is a nonparametric model, the running time grows linearly with $L$ and the computational complexity is simply $O(L)$ where $L$ is the number of ensemble members. We have added Table 5 comparing the running time of DC, S2M2_R, and DeepVoro(++), as follows:
>
> |  Methods   | #ensemble members | Time (min) |
> | ---------- | ----------------  | ---------- |
> | DC         | 1                 | 88.29      |
> | S2M2_R     | 1                 | 33.89      |
> | DeepVoro   | 1                 | 0.05       |
> | DeepVoro   | 512               | 25.67      |
> | DeepVoro++ | 1                 | 0.14       |
> | DeepVoro++ | 1280              | 179.05     |
>
> As expected, a single VD is extremely fast, $1766{\times}$ faster than DC. Even with 1280 ensemble VDs (DeepVoro++), the running time is still acceptable, $2{\times}$ slower than DC.

---

> > ### Comment · Reviewer_Qc9m · 2021-11-28
> > **Response**
> >
> > Thank you for addressing the concerns. There is a little question in A2. Dose the time represent the inference time of an n-way k-shot task? Which dataset is used for evaluation? It seems the inference speed of DC is too slow compared with other methods, such as MAML or Prototypical Net.

---

> > > ### Author Response · Authors · 2021-11-28
> > > **Thanks for your question**
> > >
> > > Thanks for your question! The time represents the inference time of *2000* 5-way 5-shot tasks on the mini-Imagenet dataset (that used to generate the main results in Table 3). Three datasets have the same N, K, and feature dimension so their times are the same. Here DC takes more time because (1) it samples thousands of new feature points for one task and (2) it trains a Logistic Regression model using scikit-learn that runs on CPU only.

---

> > > > ### Comment · Reviewer_Qc9m · 2021-11-29
> > > > **Response**
> > > >
> > > > Thank you. Please add these details or explanations for the time cost table.
> > > > All concerns have been addressed. The reviewer keeps the rating and recommends accepting this paper.

---

### Official Review · Reviewer_bCqa · 2021-11-02

**Correctness:** 4
**Technical Novelty And Significance:** 4
**Empirical Novelty And Significance:** 4
**Recommendation:** 8
**Confidence:** 4

**Details Of Ethics Concerns:**

No ethical concerns.

**Main Review:**

Strengths
- Mathematical formulation of FS approaches in terms of VD terminology
- Link to CIVD which allows ensembling different FS classifiers by means of a means of an influence functions that allows multistep
- Extensive quantitative results on mini-ImageNet, CUB, tiered_ImageNet
- Paper is well-written

Weaknesses
- Mathematical derivation of FS as a tessellation is provided for nearest neighbor and linear classifiers, it would be helpful to understand if all FS classifiers have a VD explanation, or there are some classes of FS approaches that are not VD
- The methodology section could be clarified if it has sections named after different contributions (for easy access)
- Lack of explanation of DeepVoro and DeepVoro++
- Lack of implementation details section
- Lack of discussion how the ensemble approach using CIVD is related to meta-learning techniques

Small correction:
- Typo: “To resolve this issie” -> To resolve this issue

Overall, the paper presents a mathematical link between VD and FS approaches, and shows how to use a multi-set VD diagram (CIVD) to unify multiple FS classifiers together. It is well written, and I found it easy to understand the main points of the paper. My main concern is the lack of details provided about the implementation, as I can see this approach would be useful in many future FS research techniques. It would also be valuable to add a discussion of how outliers would affect the method, computational complexity of the proposed technique. Can it be used for other tasks, such as segmentation?


**Summary Of The Paper:**

The paper presents an approach for few-shot (FS) learning using Voronoi Diagrams (VD). In particular, it relates the objectives of existing FS approaches to VD, and shows how Cluster-induced Voronoi Diagrams (CIVD), a variant of VD that allows multiple centers in a cell, can be used in for FS learning ensemble method (DeepVoro). Extensive quantitative evaluations show improvements over prior work  on three public datasets.


**Summary Of The Review:**

Mathematical link is presented between FS classification and voronoi tessellation, description of a novel ensemble FS method based on  CIVD. Paper is well written and conclusions are throughly validated. Paper needs clarifications about implementation details, and discussion of limitations, effect of outliers and generalization to other tasks.

---

> ### Author Response · Authors · 2021-11-19
> **Response to Reviewer bCqa**
>
> Thanks to Reviewer bCqa for the constructive comments! We address the concerns as follows.
>
> **Q1:** Do all FS classifiers have a VD explanation? \
> **A1:** Not all FS classifiers have a VD explanation, for example, the tree-based classifiers on the extracted features. For the FS classifiers that have a VD explanation, we have added Table 1 summarizing popular FSL methods and their underlying geometric structures.
>
> **Q2:** In Methodology, It can be clearer if the sections are named after contributions. \
> **A2:** Thanks for your suggestion. We have accordingly revised the Methods, Experiments, and Appendix so that DeepVoro--\DeepVoro\DeepVoro++ are clearly indicated in the subheadings.
>
> **Q3:** Explanation of DeepVoro and DeepVoro++. \
> **A3:** In the submitted version, we didn't mention the description of DeepVoro/DeepVoro++ until the experiment section, sorry if this causes confusion. Basically, DeepVoro is our method that uses Cluster-to-cluster Voronoi Diagram (CCVD) as the underlying geometric structure, which is then promoted to DeepVoro++ when surrogate representation is also incorporated. In the revised version, we have highlighted the definitions of DeepVoro--/DeepVoro/DeepVoro++ in Introduction, Methodology, and Experiments (see red text). Furthermore, we have added Table 1 and Table A.2 that give a clear comparison of different methods along with their geometric structures.
>
> **Q4:** Implementation details. \
> **A4:** Due to the page limit, we place the implementation details for DeepVoro-- in Sec. E.1 (page 16), for DeepVoro in Sec. F.1 (page 17), and for DeepVoro++ in Sec. G.1 (page 26).
>
> **Q5:** How the ensemble approach using CIVD is related to meta-learning? \
> **A5:** We work on transfer learning-based methods mainly because they usually outperform meta learning-based approaches for FSL (see a recent large-scale comparative study[1]). But the techniques used here (CIVD-based ensemble, surrogate representation, multi-level heterogeneities, etc.) could be easily adapted to meta learning-based FSL methods. We would leave this for future direction and we have updated the Conclusion section to include discussion of future work.
>
> **Q6:** A typo. \
> **A6:** Thanks for the correction, and we have fixed this typo (see green text).
>
> **Q7:** How outliers would affect the method? \
> **A7:** We first define outliers as the true positive samples which are out-of-distribution. It can be difficult to directly identify "outlier" in a few-shot support set. For example, in 5-way 1-shot FSL, the single-shot example itself could be an outlier. Even with 5-shot samples, it is still hard to estimate the ground-truth distribution and identify those out-of-distribution ones. To circumvent this issue, we here define Geometric Variance ($GV$) as a reflection of the possibility that a support set contains an outlier. Formally, for a support set $S = \\{(z_i, y_i)\\}\_{i = 1}^{K \times N}$, its Geometric Variance is defined as $GV(S) = \frac{1}{K} \sum_{k=1}^{K} \frac{1}{\binom N2} \sum_{i \in \{1,...,N\}, j \in \{1,...,N\}} ||z_i - z_j||_2$, measuring the average point-to-point distance in this support set. The larger $GV$ is, with higher probability $S$ contains an outlier. For every episode in 2000 episodes from 5-way 5-shot mini-ImageNet data, $GV$ and the episode accuracy are computed. Typically, very high $GV$ causes a significant decrease in episode accuracy, but our method DeepVoro is more resistant to the presence of outliers. We have added Figure K.16 for this outlier analysis.
>
> **Q8:** Computational complexity. \
> **A8:** Since an individual VD is a nonparametric model, the running time grows linearly with $L$ and the computational complexity is $O(L)$ where $L$ is the number of ensemble members. We have added Table 5 comparing the running time of DC, S2M2_R, and DeepVoro(++), as follows:
>
> |  Methods   | #ensemble members | Time (min) |
> | ---------- | ----------------  | ---------- |
> | DC         | 1                 | 88.29      |
> | S2M2_R     | 1                 | 33.89      |
> | DeepVoro   | 1                 | 0.05       |
> | DeepVoro   | 512               | 25.67      |
> | DeepVoro++ | 1                 | 0.14       |
> | DeepVoro++ | 1280              | 179.05     |
>
> As expected, a single VD is extremely fast, $1766{\times}$ faster than DC. Even with 1280 ensemble VDs (DeepVoro++), the running time is $2{\times}$ slower than DC, still acceptable.
>
> **Q9:** Can it be used for other tasks, such as segmentation? \
> **A9:** Yes, there is no obstacle to applying our method to segmentation, because segmentation is a pixel-wise classification task. But few-shot segmentation may require some specialized techniques, and we leave this for future studies.
>
> **References** \
> [1] Comparing Transfer and Meta Learning Approaches on a Unified Few-Shot Classification Benchmark, arxiv 2021.

---

> > ### Comment · Reviewer_bCqa · 2021-11-23
> > **Thanks for corrections!**
> >
> > Thank you for addressing my concerns! Analysis of runtime and outliers are particularly helpful, as it would help judge when the problem can be applied in practice.
> >
> > For the analysis of outliers, are results reported for DeepVoro or DeepVoro++? Would the choice of method affect performance?

---

> > > ### Author Response · Authors · 2021-11-24
> > > **Outliers and method selection**
> > >
> > > The outlier analysis is performed for DeepVoro on 5-shot mini-ImageNet data. Here is the rationale of choosing DeepVoro and 5-shot data for this analysis: (1) in a 1-shot support set, if the single sample itself is an outlier/noise, an FSL method may fail to learn only from outliers, so we use 5-shot data; (2) for 5-shot data, it's evident from Table 3 that DeepVoro is always better than DeepVoro++, so we select DeepVoro for this analysis.
> > >
> > > Hence the answer is yes, the choice of method will indeed affect performance. The practical guideline is to choose DeepVoro++ for one-shot learning and to choose DeepVoro for multi-shot learning. We will add this discussion to the next version. Thanks again for your constructive suggestions which have helped improve the paper.

---

### Official Review · Reviewer_JRwP · 2021-11-03

**Correctness:** 4
**Technical Novelty And Significance:** 2
**Empirical Novelty And Significance:** 2
**Recommendation:** 6
**Confidence:** 2

**Main Review:**

Positive:
- The use of the Cluster-induced Voronoi Diagram and its variant introduced in this paper are novel to FSL (to the best of my knowledge).
- The resulting voronoi-diagram formulation is geometrically elegant, and allows the integration of various classifiers, heterogeneities and types of feature representations.
- The results appear to be state of the art (at least relative to the compared works) and standard datasets used by the community.
- The ablation study appears to be a good first order approximation of what I'd expect the authors to verify.

Negative:
- The paper is a bit difficult to read. I'd attribute this primarily to the extended use of the appendix. I agree that proofs can be included primarily in the appendix, but I do not particularly appreciate that much of the ablation results and tables are included only in the appendix.
- Beyond the geometric point of view of FSL (which I do appreciate), to my understanding, the paper proposes a relatively straightforward aggregation of method/features/representations for FSL. Furthermore, the classifiers it integrates and the various heterogeneities are not novel, and neither is the cluster-induced Voronoi Diagram (which itself is a relatively straightforward generalization of voronoi diagrams).

**Summary Of The Paper:**

The authors introduce the use of Cluster-induced Voronoi Diagrams to few shot classification, and show that it can be used to combine feature and surrogate representations with various types of few-shot classifiers (eg nearest neighbour, linear classifier and cosine) and various types of heterogeneities (eg feature level, transformation level and geometry level) into a single, coherent, mathematical formulation.

**Summary Of The Review:**

I overall like the paper and appreciate the geometric formalism and results. That being said, from a mathematical point of view, I am unsure about the amount of novelty the paper would have had, had the same approaches for integration been used without the overarching geometrical point of view.

---

> ### Author Response · Authors · 2021-11-19
> **Response to Reviewer JRwP**
>
> Thanks to Reviewer JRwP for helping improve our paper! We address the concerns as follows.
>
> **Q1:** Ablation results and tables should not be included only in the appendix. Readability could be improved. \
> **A1:** Thanks for your suggestion. We have added Table 4 summarizing ablation results of DeepVoro--/DeepVoro/DeepVoro++ with different levels of ensemble. \
> To improve the readability, we have simplified the geometric terms used in the paper, highlighted the definitions of DeepVoro--/DeepVoro/DeepVoro++ in the subheadings, and added three additional tables into the main text (Table 1: method comparison, Table 2: datasets, and Table 5: running time).
>
> **Q2:** The paper proposes a method for the aggregation of various FSL classification models, but an individual model is not novel and not CIVD. \
> **A2:** The method, you mentioned, for the aggregation of various FSL classifiers is exactly CIVD/CCVD. Besides the CIVD/CCVD aggregation, other novelties include Voronoi reduction (in Sec. 3.2), compositional feature transformation (in Sec. 3.3), and surrogate representation (in Sec. 3.4). These can all be used for an individual model and are novel. You are right that an individual model (VD) is not CIVD. This is because CIVD is a flexible framework, depending on how you define the cluster. Hence, in this paper, we explore various ways of forming a cluster within which the centers can be derived from VD/PD (DeepVoro--), different data augmentations/feature transformations (DeepVoro), or feature/surrogate representations (DeepVoro++). Please see two new tables Table A.1 and A.2 for how different choices of defining the center lead to different geometric models.
>
> **Q3:** Unsure about the novelty of the paper, if the same approach for integration has been used for FSL. \
> **A3:** To the best of our knowledge, our work is the first one that utilizes CIVD for this integration. Other integration methods significantly differ from ours, for example, the ensemble in [1] is restricted by the number of networks which is typically ${<}20$ due to the high computation load of retraining deep models; whereas in [2], ensemble consists of models learned at each epoch, which may potentially limit the diversity of ensemble members. Our method enables efficient ensemble of ${>}1000$ individual models (VDs) without the need of pre-training the deep network. \
> The novelties of the paper could be summarized as follows: \
> <1> The first work that reveals the underlying geometric structures for a number of prevailing FSL methods. \
> <2> Proposes to use more advanced geometric structures (CIVD and CCVD) for better FSL (DeepVoro). \
> <3> Proposes a novel surrogate representation that significantly improves 1-shot FSL (DeepVoro++). \
> <4> Achieved state-of-the-art results on all three benchmark datasets.
>
> **References** \
> [1] Diversity with cooperation: Ensemble methods for few-shot classification, CVPR 2019. \
> [2] An ensemble of epoch-wise empirical bayes for few-shot learning, ECCV 2020.

---

> > ### Comment · Reviewer_JRwP · 2021-11-25
> > **Response**
> >
> > Thank you for the changes -- I think they should improve readability.
> > I'm happy to maintain my rating, and I believe the paper should be accepted.

---

### Author Response · Authors · 2021-11-19
**Summary of Revisions**

We thank AC for handling this paper, thank Reviewers JRwP (R1), bCqa (R2), Qc9m (R3), and uC53 (R4) for acknowledging the novelty and contributions of the paper, and for raising concerns regarding the readability and presentation. We take into consideration all the comments and systematically revise the paper. The major changes can be summarized as follows:

1. **Notations** (R2-Q3, R4-Q7). We have highlighted the definitions of DeepVoro--/DeepVoro/DeepVoro++ in Introduction, Methodology, and Experiments (see red text). Furthermore, we have added Table 1 and Table A.2 that give a clear comparison of different methods along with their geometric structures.
2. **Ablation Study** (R1-Q1, R3-Q1, R4-Q11). We have added Table 4 showing DeepVoro--/DeepVoro/DeepVoro++'s performance with varied levels of ensemble.
3. **Terminology and readability** (R4-Q5). To improve the readability, we have simplified the geometric terms used in the paper, and accordingly, modified the title to "**Few-shot Learning as Cluster-induced Voronoi Diagrams: A Geometric Approach**". To improve the self-containment and readability of the paper, we have added Table A.1 summarizing and comparing VD, PD, CIVD, and CCVD, four geometric structures discussed in the paper. We hope these would make the paper easier to understand especially for readers unfamiliar with computational geometry.
4. **Running time** (R2-Q8, R3-Q2). We have added Table 5 comparing the running time of DeepVoro(--/++) with other methods, demonstrating the efficiency of our method.
5. **Outlier analysis** (R2-Q7). We have added Figure K.16 illustrating that our method is more robust to outliers than other methods.

We hope that our responses and revisions address all reviewers’ concerns. Please let us know and we are more than happy to answer any further questions.

---

### Decision · Program_Chairs · 2022-01-20

**Decision:**

Accept (Poster)

**Comment:**

This paper analyzes popular metric-based few-shot learning (FSL) methods from the perspective of computational geometry. Namely, viewing prototypical networks as Voronoi diagrams (VDs). This lends itself to incorporating extensions based on the recently proposed CIVD that allows for multiple centers per cell. The paper then discusses various aspects of the FSL pipeline (data augmentation, feature transformations, geometries and representations), referred to as heterogeneities, that can be efficiently ensembled via a cluster-to-cluster VD (CCVD). The resulting model produces state-of-the-art results.

Initial concerns from the reviewers pointed to a potential lack of novelty (since it can be seen as applying existing ideas to FSL), lack of self-containment in the main paper, weak positioning in the context of other FSL methods (which ones can be interpreted under the VD framework) and a potentially impractical computational complexity. The discussion period settled these issues, with the paper receiving several updates, and the reviewers all ended up recommending acceptance.

Personally, I would like to see an addition to Figure 1 with the resulting decision boundaries from CIVD and CCVD. I think that this would greatly improve the ability of the reader to reason about the approach intuitively. Also, as a minor comment, I think that the argmax below eqs 1 and 7 should either be an argmin, or the distances should be negated. Otherwise, I think this is a valuable contribution to the FSL literature.